# Fatal COVID-19 pulmonary disease involves ferroptosis

Baiyu Qiu[1,11], Fereshteh Zandkarimi [1,2,11], Anjali Saqi[3,11], Candace Castagna [4], Hui Tan[1], Miroslav Sekulic[3], Lisa Miorin[5,6], Hanina Hibshoosh[3], Shinya Toyokuni [7,8], Koji Uchida[9] & Brent R. Stockwell [1,3,10] ✉

SARS-CoV-2 infection causes severe pulmonary manifestations, with poorly understood mechanisms and limited treatment options. Hyperferritinemia and disrupted lung iron homeostasis in COVID-19 patients imply that ferroptosis, an iron-dependent cell death, may occur. Immunostaining and lipidomic analysis in COVID-19 lung autopsies reveal increases in ferroptosis markers, including transferrin receptor 1 and malondialdehyde accumulation in fatal cases. COVID-19 lungs display dysregulation of lipids involved in metabolism and ferroptosis. We find increased ferritin light chain associated with severe COVID-19 lung pathology. Iron overload promotes ferroptosis in both primary cells and cancerous lung epithelial cells. In addition, ferroptosis markers strongly correlate with lung injury severity in a COVID-19 lung disease model using male Syrian hamsters. These results reveal a role for ferroptosis in COVID-19 pulmonary disease; pharmacological ferroptosis inhibition may serve as an adjuvant therapy to prevent lung damage during SARS-CoV-2 infection.

Coronavirus disease-2019 (COVID-19), caused by infection with the coronavirus SARS-CoV-2, leads to pulmonary manifestations of pneumonia and acute respiratory distress syndrome (ARDS). The latter accounts for the poor prognosis and high mortality in severe cases of the disease. Acute lung injury (ALI), including diffuse alveolar damage (DAD), is the histological correlate of ARDS; its early stage is defined by the presence of hyaline membranes, edema, and fibrosis[1]. Other patterns of non-acute lung injury (non-ALI), such as pulmonary vascular congestion with hemangiomatosis-like changes and microthrombi is another common pathology in COVID-19 patients[2]. COVID-19 pulmonary pathology is associated with both viral infection damage and host inflammatory responses, including cytokine storm and inflammation[3]. Chronic inflammatory responses by neutrophils and macrophages may aggravate pulmonary tissue damage. Infiltrating immune cells release reactive oxygen species and free radicals, causing oxidative injury. Although inflammation is associated with lethal COVID-19 lung pathologies, the pathogenic mechanisms of pulmonary manifestations remain unclear.

The lung is the most affected organ during SARS-CoV-2 infection, and there is currently no curative treatment for COVID-19 lung disease. Supportive treatments, such as intubation and mechanical ventilation, are utilized to enable oxygen inhalation in patients experiencing respiratory failure[4]. The current treatment for COVID-19 ARDS is a combination of anti-viral and anti-inflammatory medications. Protease

[1]Department of Chemistry, Columbia University, New York, NY 10027, USA. [2]Mass Spectrometry Core Facility, Department of Chemistry, Columbia University, New York, NY 10027, USA. [3]Department of Pathology and Cell Biology, Columbia University Irving Medical Center, New York, NY 10032, USA. [4]Institute of Comparative Medicine, Columbia University Irving Medical Center, New York, NY 10032, USA. [5]Department of Microbiology, Icahn School of Medicine at Mount Sinai, New York, NY 10029, USA. [6]Global Health Emerging Pathogens Institute, Icahn School of Medicine at Mount Sinai, New York, NY 10029, USA. [7]Department of Pathology and Biological Responses, Nagoya University Graduate School of Medicine, Nagoya 466-8550, Japan. [8]Center for Low-temperature Plasma Sciences, Nagoya University, Furo-Cho, Chikusa-ku Nagoya 464-8603, Japan. [9]Graduate School of Agricultural and Life Sciences, The University of Tokyo, Tokyo 113-8657, Japan. [10]Department of Biological Sciences, Columbia University, New York, NY 10027, USA. [11]These authors contributed equally: Baiyu Qiu, Fereshteh Zandkarimi, Anjali Saqi. ✉e-mail: bstockwell@columbia.edu

inhibitors have been used for treating early viral infection to inhibit SARS-CoV-2 replication[5]. Immunomodulatory drugs such as corticosteroids and IL-6 receptor inhibitors have shown benefits in survival and reducing mechanical ventilation in critically ill patients[6]. Other therapeutic strategies that have been proposed for targeting specific clinical pathologies include anti-coagulant treatment and mesenchymal stem cell therapy[7,8]. Due to the high demand for effective treatments for COVID-19 pulmonary complications, novel therapeutic strategies and understanding of pathogenesis remain crucial.

Ferroptosis is an iron-dependent, phospholipid-peroxidation-driven form of non-apoptotic cell death that is involved in a variety of disease processes[9]. Ferroptosis is characterized by extensive peroxidation of phospholipids containing polyunsaturated fatty acyl tails (PL-PUFAs). Lipid peroxidation overwhelms cellular repair systems, such as the glutathione peroxidase 4 (GPX4) pathway, the ferroptosis suppressor protein 1 (FSP1) pathway, and the GTP cyclohydrolase 1 (GCH1) pathway, leading to cell death[10].

Pro-ferroptosis molecular changes are associated with SARS-CoV-2 infection and may trigger ferroptosis in the lung. Firstly, GPX4 expression is decreased in SARS-CoV-2-infected Vero cells, which increases susceptibility to lipid peroxidation by reducing lipid repair capability[11]. Secondly, the SARS-CoV-2 structural protein inhibits the NRF2 antioxidant response pathway, which normally protects against oxidative damage[12]. Thirdly, serum ferritin is elevated in critically ill COVID-19 patients and correlates with disease severity[13,14]; serum ferritin supplies labile iron that drives ferroptosis. Altered expression of iron homeostasis proteins and accumulation of reactive iron reflect the disruption of iron metabolism in the lung[15]. Fourthly, lipid peroxidation was reported during cardiovascular complications in a COVID-19 patient and in primary pacemakers of the heart in a SARS-CoV-2-infected hamster model[16,17]. Thus, ferroptosis may contribute to pulmonary tissue damage in COVID-19 patients.

Here, we identified ferroptosis as the major cell death mechanism that underlies COVID-19 lung disease. We investigated cell death markers on human COVID-19 lung tissue and found distinct molecular features of ferroptosis in severe lung pathologies, including ALI and non-ALI pathologies. We analyzed the lipid profile of COVID-19 patient lung autopsies using mass-spectrometry-based lipidomics and found lipid changes involving inflammation, metabolism, and ferroptosis. We further analyzed a COVID-19 lung disease model using Syrian hamsters and found correlation of ferroptosis with lung pathology. These findings highlight a critical role for ferroptosis in promoting COVID-19 lung pathology and suggest that suppressing this iron-dependent cell death may be beneficial.

## Results

### Ferroptosis is elevated in post-mortem COVID-19 lungs

The major COVID-19 pulmonary pathologies include acute lung injury (ALI), featuring diffuse alveolar damage (DAD), and non-ALI, featuring congestion and microthrombi[2]. To investigate COVID-19 pulmonary disease across different pathological groups, we obtained severe COVID-19 patient lung tissue, including ALI and non-ALI cases. Under microscopic examination, the ALI samples were identified by DAD with characteristic hyaline membrane deposition lining the alveolar septa (Fig. 1a). The non-ALI samples were identified by lung parenchyma with vascular congestion and hemangiomatosis-like changes exhibiting a mesh-like framework on the alveolar walls. For the non-affected control group, we obtained healthy lung sections selected from uninvolved areas of pre-pandemic pneumothorax lung resections and neoplasm lung resections. We also obtained a non-COVID-19 ALI control group using lung autopsy samples from patients deceased from ARDS before the pandemic. All control lungs lacked SARS-CoV-2 infection.

To define the cell death mechanisms associated with COVID-19 lung pathology, we performed immunofluorescence and immunohistochemistry staining on human lung samples with cell death markers specific for ferroptosis, necroptosis, apoptosis, and pyroptosis. For ferroptosis detection, we evaluated two previously established ferroptosis markers, transferrin receptor 1 (TfR1) and malondialdehyde adduct (MDA)[18,19]. We found that TfR1 and MDA were elevated in ALI and non-ALI lungs compared with control lungs, suggesting the occurrence of ferroptosis (Fig. 1b–e). Non-COVID-19 ALI cases did not show increased TfR1, suggesting that ferroptosis activation is specific to SARS-CoV-2 infection (Fig. 1d).

To evaluate the association of other forms of cell death, we measured other cell death markers, including a necroptosis-specific marker, phosphorylated mixed lineage kinase domain like pseudokinase (phospho-MLKL); an apoptosis-specific marker, cleaved Caspase 3; and a pyroptosis-specific marker, cleaved Gasdermin D[20–22]. We did not detect an increase in these markers in either ALI or non-ALI lungs in comparison with control lungs (Fig. 1f, Supplementary Fig. 1c). These results indicate that ferroptosis is the major cell death mechanism detected in fatal COVID-19 lung samples.

To validate the specificity of these cell death markers, we evaluated positive and negative controls in tissues and cell lines. For the ferroptosis markers, we evaluated tissue from diffuse large B cell lymphoma (DLBCL, SUDHL6) xenografted mice treated with the ferroptosis-inducing compound IKE or vehicle[23]. The ferroptosis positive control showed significantly higher staining for both TfR1 and MDA antibodies, compared with the vehicle control (Supplementary Fig. 1a). No cross-staining with phospho-MLKL antibody was observed in the ferroptosis positive control. For the necroptosis and apoptosis markers, we stimulated cell death in cell lines using necroptosis-specific and apoptosis-specific inducers. Both phospho-MLKL and cleaved Caspase 3 antibodies had significantly increased staining in their respective positive control samples compared with vehicle control (Supplementary Fig. 1b). Overall, these results confirmed the consistency of these antibodies used in the study and reinforce the conclusion that ferroptosis is the mode of cell death triggered in the lungs of patients with fatal COVID-19 disease.

We also evaluated the abundance of the key cell stress and cell death regulators, the tumor suppressor p53 and the E3 ligase MDM2 in COVID-19 lungs. MDM2 has been shown to promote ferroptosis through lipid remodeling, and p53 has been shown to promote ferroptosis as part of its tumor suppression mechanism[24,25]. We found that p53 and MDM2 were significantly reduced in the ALI group, while no change was observed in the non-ALI group (Supplementary Fig. 1d). The lack of elevation of p53 or MDM2 suggests that these proteins do not drive the enhanced ferroptosis detected in COVID-19 lung tissues.

### Dysregulation of iron homeostasis contributes to ferroptosis

To investigate whether hyperferritinemia is associated with severe COVID-19 lung disease, we obtained the serum ferritin records of 12 subjects involved in this study. We compared the initial and final serum ferritin measurements of patients during their hospitalization period. All COVID-19 patients experienced elevation of serum ferritin during hospitalization and the majority displayed levels above the normal range (Fig. 2a, Supplementary Data 1). These data suggest that elevated serum ferritin is a common feature of patients with fatal COVID-19.

We hypothesize that iron overload is associated with elevated ferroptosis in infected lungs. We evaluated the expression of TfR1 and ferritin light chain (FTL) in COVID-19 lungs with severe and mild pathology, obtained from lung explant/biopsy of severe and recovered COVID-19 patients, respectively. We found a significant elevation of both iron regulators in severe cases but not in mild cases (Fig. 2b, Supplementary Fig. 2a). TfR1 is both a ferroptosis-specific marker and is involved in controlling iron homeostasis through the transport of holo-transferrin that contributes to intracellular labile iron pool[26]. Ferritin is an iron-storage protein that contributes to the labile iron

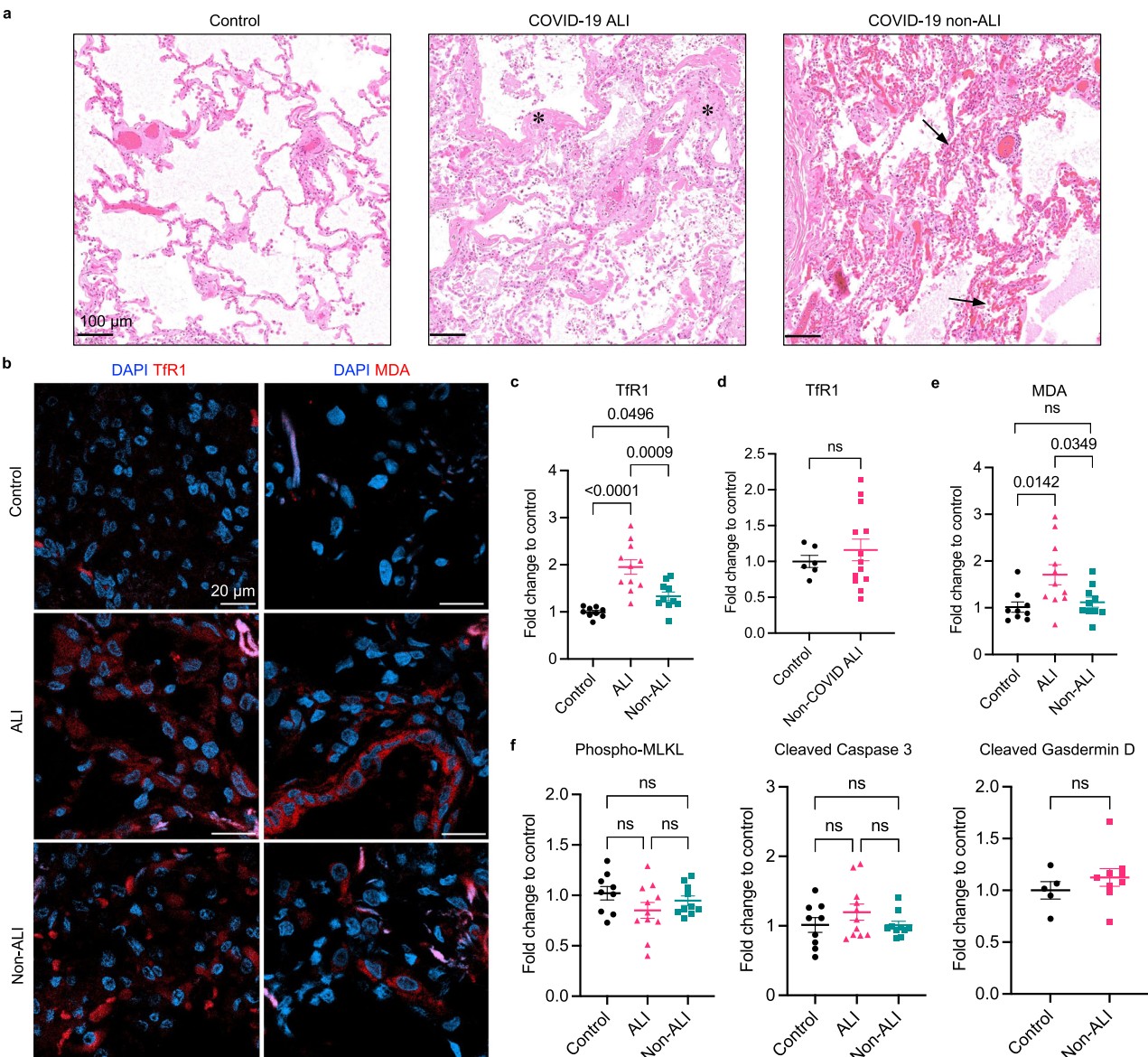

**Fig. 1 | Ferroptosis is elevated in post-mortem COVID-19 lungs. a** Representative images of H&E-stained COVID-19 lung autopsies with ALI and non-ALI pathology and non-infected control lungs. ALI case shows characteristic hyaline membranes lining the alveolar walls (asterisks). Non-ALI case shows congestion and hemangiomatosis-like changes in the alveolar wall (arrows). Scale bar = 100 µm. **b** Representative images of immunofluorescence (IF) staining using anti-TfR1 antibody (clone 3F3-FMA) and anti-MDA antibody (clone 1F83). Nuclei are shown in blue, and antibodies are shown in red. Scale bar = 20 µm. **c** The mean intensity of TfR1 signal of each case is normalized to the mean of non-infected control group. Data shown as mean ± SEM, $n = 9$ (control), $n = 11$ (ALI), $n = 10$ (non-ALI), one-way ANOVA ($p$ value indicated). **d** Non-COVID-19 ALI cases were immunohistochemistry

(IHC) stained with anti-TfR1 antibody (clone H68.4). Positive stain area is normalized to control group. Data shown as mean ± SEM, $n = 6$ (control), $n = 13$ (non-COVID), unpaired two-sided $t$ test. **e** The mean intensity of MDA signal is normalized to the non-infected control group. Data shown as mean ± SEM, $n = 9$ (control), $n = 11$ (ALI), $n = 10$ (non-ALI), one-way ANOVA ($p$ value indicated). **f** COVID-19 and control cases were stained with anti-phospho-MLKL, anti-cleaved Caspase 3, and anti-cleaved Gasdermin D antibodies. The mean intensity of each antibody is normalized to the control group. Data shown as mean ± SEM, $n = 9$ (control), $n = 11$ (ALI), $n = 10$ (non-ALI) (left and middle panel), one-way ANOVA. $n = 5$ (control), $n = 10$ (non-ALI) (right panel), unpaired two-sided $t$ test.

pool. The elevation of both iron regulatory proteins is closely related to iron overload.

To investigate the heterogeneity of gene expression across different alveolar cell types, we analyzed a single-cell RNA sequencing dataset (GEO, GSE171524) on COVID-19 post-mortem lungs obtained from the same patient cohort used in our study[27]. We analyzed the transcription of 9 major iron regulatory genes, among which FTL, ferritin heavy chain 1 (FTH1), and TfR1 are the most elevated genes in COVID-19 lung (Fig. 2c). A UMAP plot shows that FTL and TfR1 expression are elevated primarily in macrophages, fibroblasts, and other epithelial cells (Fig. 2d). Despite the high basal level of FTL in

macrophages, we found FTL signals outside of macrophage markers (CD68 and CD11c) in severe COVID-19 lungs, suggesting that other alveolar cell types were involved in FTL upregulation (Supplementary Fig. 2b). In addition, the expression of the anti-ferroptosis genes GPX4 and FSP1 was not significantly changed in COVID-19 lungs (Supplementary Fig. 2c, d).

To evaluate iron dysregulation and ferroptosis in alveolar cells, we generated an iron-overload cell model by treating human primary lung epithelial cells and an epidermoid carcinoma cell line, Calu-1, with ferric ammonium citrate (FAC)[28]. FAC-induced lipid peroxidation accumulation in primary epithelial cells was measured by a fluorescent

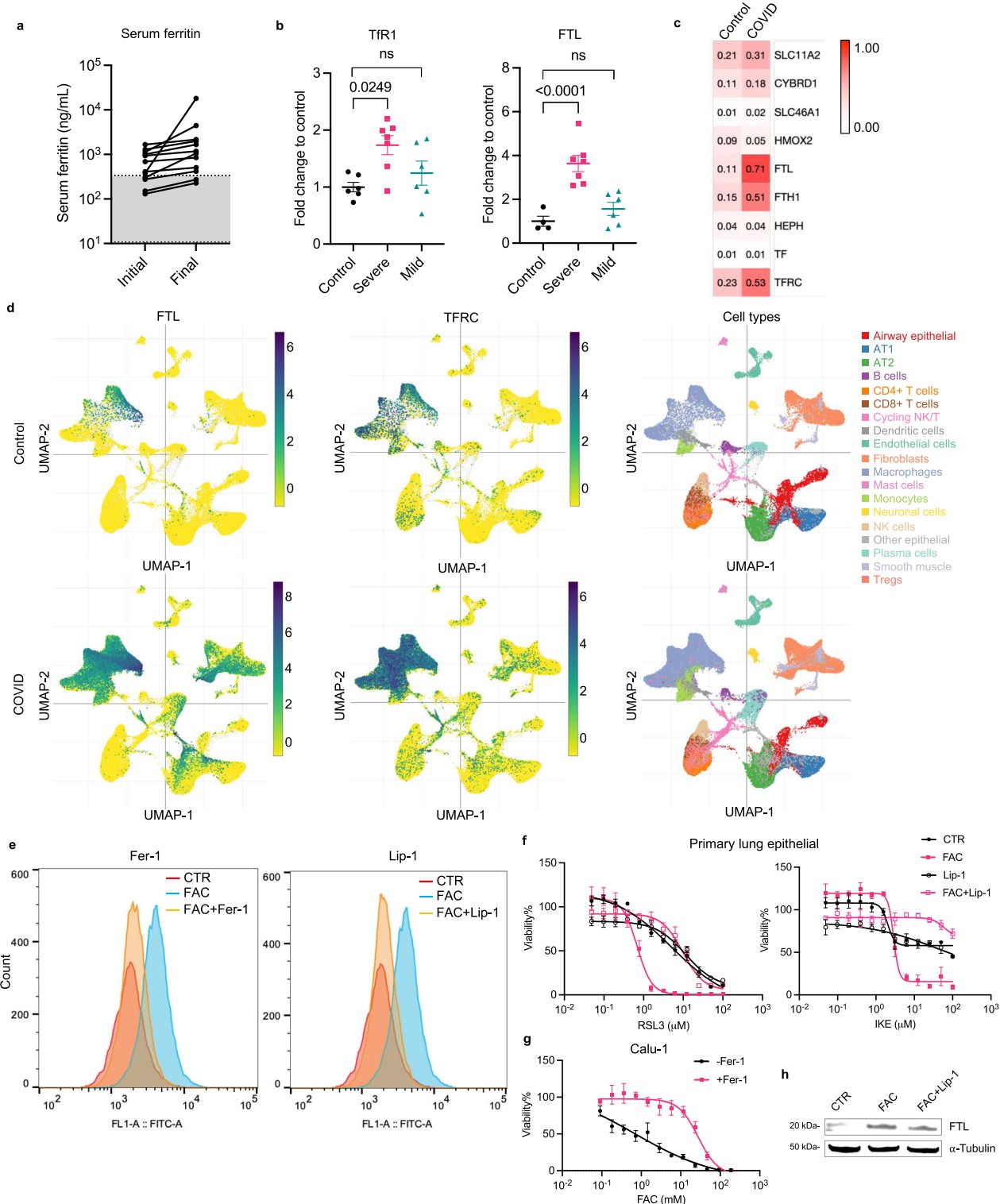

lipid probe, C11-BODIPY, and this increase was suppressed by ferroptosis inhibitors, ferrostatin-1 and liproxstatin-1 (Fig. 2e, Supplementary Fig. 5). FAC treatment alone was sufficient to induce cell death rescuable by ferrostatin-1 in Calu-1, but not in the primary epithelial cells (Fig. 2g, Supplementary Fig. 2e). Interestingly, FAC co-treatment sensitizes primary epithelial cells to both ferroptosis inducers, RSL3 and IKE (Fig. 2f)[23,29]. FAC treatment also induced FTL expression in primary epithelial cells and Calu-1, consistent with our analysis of human lung tissue (Fig. 2h, Supplementary Fig. 2h, 6a, b). An alternative iron source, hemoglobin, was evaluated but did not induce ferroptosis or

lipid peroxidation in primary epithelial cells (Supplementary Fig. 2e, g). Altogether, these data confirm that iron overload is associated with increased FTL expression and lipid peroxidation accumulation, which contributes to ferroptosis in lung cells.

## Lipidomics reveals evidence of ferroptosis in fatal COVID-19 lung samples

To investigate changes in lipid profile after SARS-CoV-2 infection, we performed mass spectrometry-based untargeted lipidomics on COVID-19 lung autopsies (ALI, $n = 11$; non-ALI, $n = 2$) and non-infected

**Fig. 2 | Dysregulation of iron homeostasis contributes to ferroptosis. a** The initial and last serum ferritin level in deceased COVID-19 patient ($n = 12$) during hospitalization. Gray area represents normal range (11–336 ng/mL). Connected dots represents the same patient. **b** Severe and mild COVID-19 explant/biopsy, and control cases were IHC stained with anti-TfR1 antibody (clone H68.4) and IF stained with anti-ferritin light chain (FTL) antibody. The positive stain area of TfR1 and mean intensity of FTL are normalized to the non-infected control group. Data shown as mean ± SEM, $n = 4$ (control), $n = 7$ (severe), $n = 6$ (mild) (left panel). $n = 6$ (control), $n = 7$ (severe), $n = 6$ (mild) (right panel). One-way ANOVA ($p$ value indicated). **c** Reanalysis of single cell RNA sequencing dataset (GEO, GSE171524). Plot shows the scaling relative to each gene's expression across all cells associated with each column label in the plot. Iron regulatory pathway genes are shown. **d** UMAP plot shows the expression of FTL and TfR1 across different alveolar cell types in

COVID-19 vs. control groups. **e** Primary lung epithelial cells were treated with 20 mg/mL ferric ammonium citrate (FAC) with or without 10 μM ferrostatin-1 (Fer-1) or 10 μM liproxstatin-1 (Lip-1) for 5 h. Lipid peroxidation was measured by C11-BODIPY$^{581/591}$ using flow cytometry. Data shown as representative result of 2 independent experiments. **f** Primary lung epithelial cells were co-treated with 2 mg/mL FAC with or without 10 μM Lip-1, and the dose responses to RSL3 and IKE at 24 h were measured. Data shown as mean ± SD of $n = 3$ technical replicates. **g** Dose response curve of FAC in Calu-1 cells at 24 h with or without 10 μM Fer-1. Data shown as mean ± SD of $n = 3$ technical replicates. **h** Western blot analysis of primary lung epithelial cells treated with 20 mg/mL FAC with or without 10 μM Lip-1 for 5 h. Whole cell lysate was collected and 40 μg of protein was loaded to each lane. FTL and α-Tubulin were blotted. Data shown as representative result of 2 independent experiments. Uncropped blots are provided in Supplementary Fig. 6.

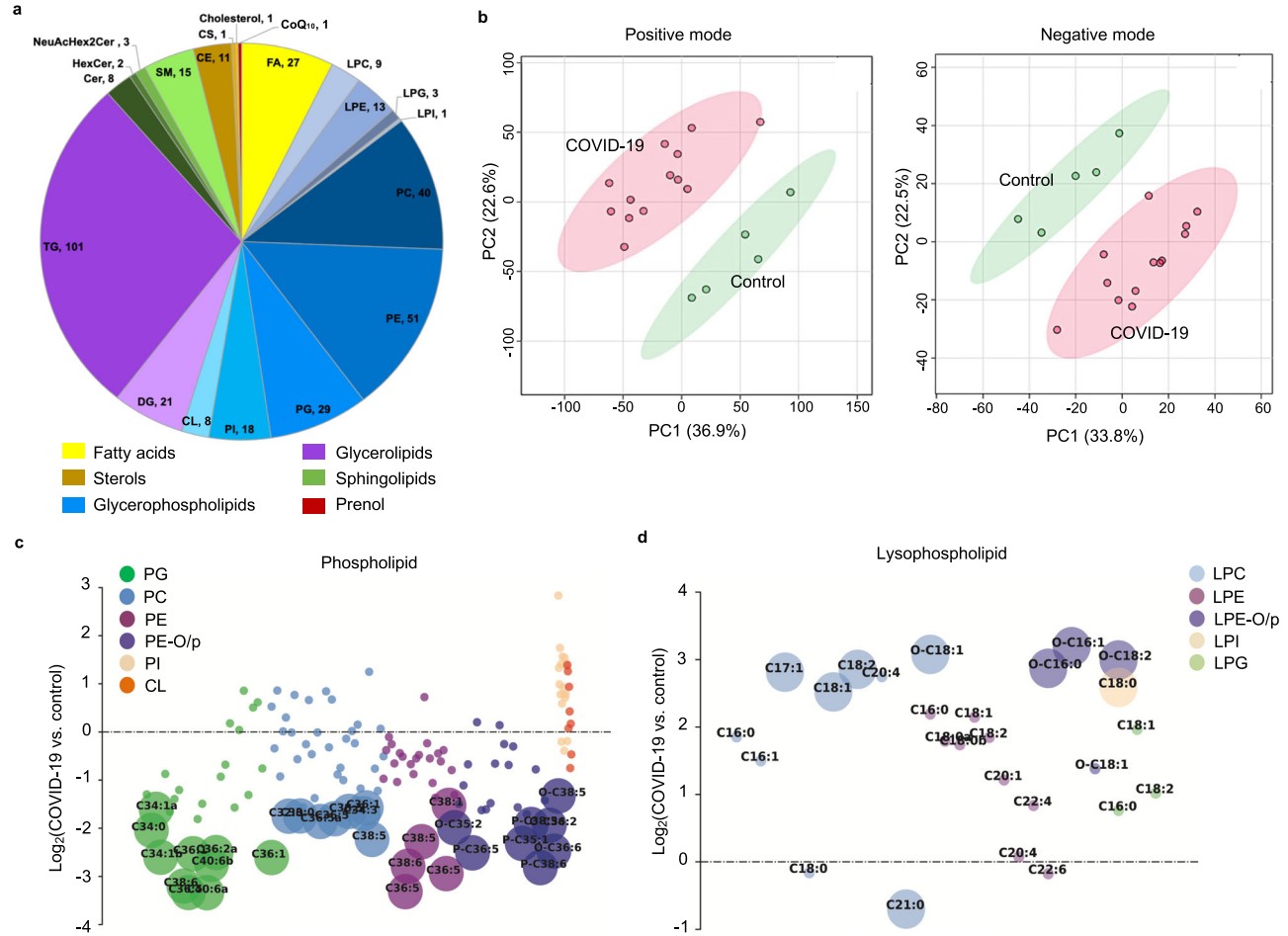

**Fig. 3 | Lipidomics reveals evidence of ferroptosis in fatal COVID-19 lung samples. a** Lipids were extracted from COVID-19 lung autopsy samples and analyzed using mass spectrometry. A total of 363 unique lipids were identified across 6 lipid categories and 20 subclasses combined in both positive and negative ESI modes. Values beside each subclass annotation represent the number of lipids identified in that particular subclass. **b** PCA plots show clear separation of COVID-19 groups ($n = 13$) from control ($n = 5$) groups in both positive and negative electrospray ionization modes. Bubble plots of log$_2$ fold changes in abundance of

identified (**c**), phospholipids and (**d**), lysophospholipids in COVID-19 lung relative to the control lung are shown. Bubble size represents the FDR-corrected $p$ value from the Welch's $t$ test. CoQ$_{10}$ coenzyme Q$_{10}$, Cer ceramide, Hex2Cer dihexosylceramide, NeuAcHex2Cer ganglioside, SM sphingomyelin, CE cholesteryl ester, CS cholesterol sulfate, FA fatty acid, LPC lyso phosphatidylcholine, PC phosphatidylcholine, LPE lysophosphatidylethanolamine, PE phosphatidylethanolamine, LPG lysophosphatidylglycerol, PG phosphatidylglycerol, LPI lysophosphatidylinositol, PI phosphatidylinositol, CL cardiolipin, DG diglyceride, TG triglyceride.

control lung tissue ($n = 5$). We identified 363 unique lipid species across 6 lipid categories and 19 subclasses in COVID-19 patient lungs, using both positive and negative electrospray ionization (ESI) modes (Fig. 3a). 130 unique lipids were significantly altered, using FDR-corrected $p < 0.05$ and fold change > 1.5, in the COVID-19 group

compared with the control group. Principal component analysis (PCA) of the extracted lipid features revealed clustering and separation of COVID-19 samples from control samples in both positive and negative ionization modes, indicating substantial lipid differences (Fig. 3b).

The COVID-19 samples were depleted in phospholipid species containing polyunsaturated fatty acyl tails (PL-PUFAs) and accumulated lysophospholipids (Fig. 3c, d, Supplementary Data 2). PL-PUFAs are susceptible to oxidation and formation of lipid hydroperoxides, causing their depletion during ferroptosis[30]. The oxidized PUFA tail then gets cleaved by phospholipases, causing accumulation of lysophospholipids (lysoPLs). Loss of PL-PUFAs and accumulation of lysoPLs are thus hallmarks of ferroptosis[31]. These lipidomic changes reflect ferroptosis in COVID-19 patient lung samples.

Among the other differentially altered lipid species, surfactant phospholipids, such as dipalmitoyl-phosphatidylcholine (DPPC; PC(16:0, 16:0)) and palmitoyl-oleoyl-PG (PG(16:0, 18:1)) were depleted in infected lungs (Supplementary Fig. 3a). Lung surfactant phospholipids have critical roles in reducing surface tension at the air-liquid interface; their depletion indicates disruption of normal lung function after viral infection[32]. We also observed significant accumulation in both monounsaturated and polyunsaturated long chain fatty acids in infected lungs (Supplementary Fig. 3b). The elevated levels of free fatty acids may be due to increased fatty acid biosynthesis, which has been suggested to promote viral replication, invasion, and fusion[33]. Finally, we found significantly increased cholesteryl esters and triglycerides— major storage lipids in COVID-19 lungs (Supplementary Fig. 3c). Accumulation of these lipids at ectopic sites impairs cellular functions, disrupts intercellular signaling, and causes lipotoxicity, inflammation, and cell death[34]. Overall, the lipidomic analysis suggests substantial lipid dysregulation after viral infection, with ferroptosis being prominent in the lungs of COVID-19 patients.

### Ferroptosis correlates with lung disease severity in a COVID-19 Syrian hamster model

To evaluate ferroptosis during the progression of COVID-19 lung pathology, we established an animal model using male Syrian hamsters that express the angiotensin-converting enzyme 2 (ACE2) receptor necessary for SARS-CoV-2 infection[35]. We intranasally inoculated 10-week-old male Syrian hamsters with the wild-type SARS-CoV-2 strain, the same as the strain found in patients recruited in this study. We examined the lung manifestations of Syrian hamsters at different time points post-infection. We detected a continuous drop in body weight in the first week post-infection, and weight rebound in the second week, consistent with other clinical symptoms, including lethargy, ruffled fur, hunched backs, and a lack of grooming which were reduced in the recovery phase (Supplementary Fig. 4a). Body temperature was not significantly affected by infection in this model. The SARS-CoV-2 nucleocapsid protein was detected in the lungs of SARS-CoV-2-infected animals, but not mock-infected animals, indicating viral replication in the lung (Supplementary Fig. 4b).

To evaluate lung injury in infected hamsters, we stained lung sections with hematoxylin and eosin and performed histopathological examination and quantification. We observed mild to severe lung injury on day 2 that gradually increased and peaked on day 7, and eventually resolved on day 14 post infection (Fig. 4a). Infected lungs exhibited COVID-19 lung pathologies, which were absent in mock-infected animals. Areas of injury involved airways (e.g., reactive rounded epithelial cells with loss of polarity), vessels (e.g., perivascular edema and chronic inflammation), and alveoli (e.g., septa thickened by minimal chronic inflammation, lined by reactive type II pneumocytes and containing airspace macrophages). We did not observe any acute or chronic pathologies in other organs, including the heart (cardiomyocytes, interstitium, and vasculature) and the kidney (glomeruli, tubules, interstitium, and vasculature) (Supplementary Fig. 7).

We evaluated ferroptosis in infected lungs by immunohistochemical staining of transferrin receptor 1 (TfR1). TfR1 was significantly increased on day 6 and day 7 post infection compared with mock-infected animals (Fig. 4b, Supplementary Fig. 4c). Expression of TfR1 correlated with severity of lung injury during both the progression

stage and the recovery stage (Fig. 4c). The lipid peroxidation end product, 4-hydroxynonenal (4-HNE), also increased in infected lungs, peaking on day seven (Fig. 4d, Supplementary Fig. 4d, e)[36]. Ferritin light chain (FTL) was significantly elevated and colocalized with 4-HNE signals in infected hamster lungs compared with mock-infected group, which is consistent with analysis on human lung samples (Fig. 4e, f, Supplementary Fig. 4d). Apoptosis was measured by terminal deoxynucleotidyl transferase dUTP nick end labeling (TUNEL); we did not detect any increases in apoptosis in infected lungs (Fig. 4a). These results confirm that ferroptosis is associated with lung pathology resulting from SARS-CoV-2 infection, consistent with our analysis of human patient tissue.

To evaluate the effect of ferroptosis inhibition on COVID-19 lung pathology, we intraperitoneally injected hamsters daily with one of two ferroptosis inhibitors, liproxstatin-1 or a ferrostatin-1 analog, Th-2-31, that showed high potency in inhibiting IKE-induced ferroptosis in HT-1080 (Supplementary Fig. 4f). Liproxstatin-1-treated animals showed ameliorated lung injury compared with vehicle-treated animals, while the effect of Th-2-31 was not significan (Supplementary Fig. 4k). Ferroptosis in the lung measured by TfR1 was not significantly reduced in the Th-2-31-treated group compared with the vehicle-treated group (Supplementary Fig. 4h). The efficacy of ferroptosis inhibitors was limited by drug accumulation in the lung and drug stability in hamster plasma (Supplementary Fig. 4g, i, j). Altogether, these data suggest a strong correlation between ferroptosis and COVID-19 lung pathology, and ferroptosis inhibition may serve as adjuvant therapy to reduce lung injury.

## Discussion

In this study, we investigated the molecular features of COVID-19-associated pulmonary pathologies in human lung autopsies and a hamster model; we detected ferroptosis as the major cell death mechanism associated with lung damage caused by SARS-CoV-2 infection, suggesting a target for therapeutics. We detected predominant elevation of ferroptosis markers, including iron dysregulation, lipid peroxidation, depletion in PL-PUFAs, and elevation of lysophospholipids in post-mortem COVID-19 lung autopsy samples. These data support a driving role of ferroptosis during COVID-19 lung disease progression.

Fatal COVID-19 lung pathology includes ALI (diffuse alveolar damage (DAD)) and non-ALI (congestion, hemangiomatosis-like change, and thrombi). We found significant elevation of ferroptosis features in both types of lung pathologies in patient autopsies. Blood accumulation in lung parenchyma is often associated with COVID-19 lung manifestations. Ferroptosis is evident in blood-leakage-associated diseases, such as intracerebral hemorrhage, caused by the rupture of neuronal vessels and the leakage of blood into interstitial tissues[37]. Dead blood cells release iron and cytotoxic agents into surrounding tissue areas that lead to increasing inflammation and tissue damage.

We observed high serum ferritin and increased ferritin light chain in the lung tissue of severe COVID-19 cases, consistent with other studies of hyperferritinemia in critically ill COVID-19 patients (Fig. 2a, b). High serum ferritin largely reflects an increase in iron pool that leads to disruption of iron homeostasis and iron toxicity. Alveolar macrophages contain high level of iron during erythrocyte recycling, which may trigger ferroptosis. We previously revealed in a different model that ferroptotic macrophages induced by asbestos secrete holo-ferritin-loaded exosomes, which are received by mesothelial cells to cause oxidative DNA damage and carcinogenesis[38]. Iron-rich extracellular vesicles secreted by ferroptotic cells, such as macrophages, spread cell-death signals to surrounding environments and exacerbate tissue damage.

In our iron-overload cell model, ferric ammonium citrate (FAC) induced lipid peroxidation in primary lung epithelial cells, which was

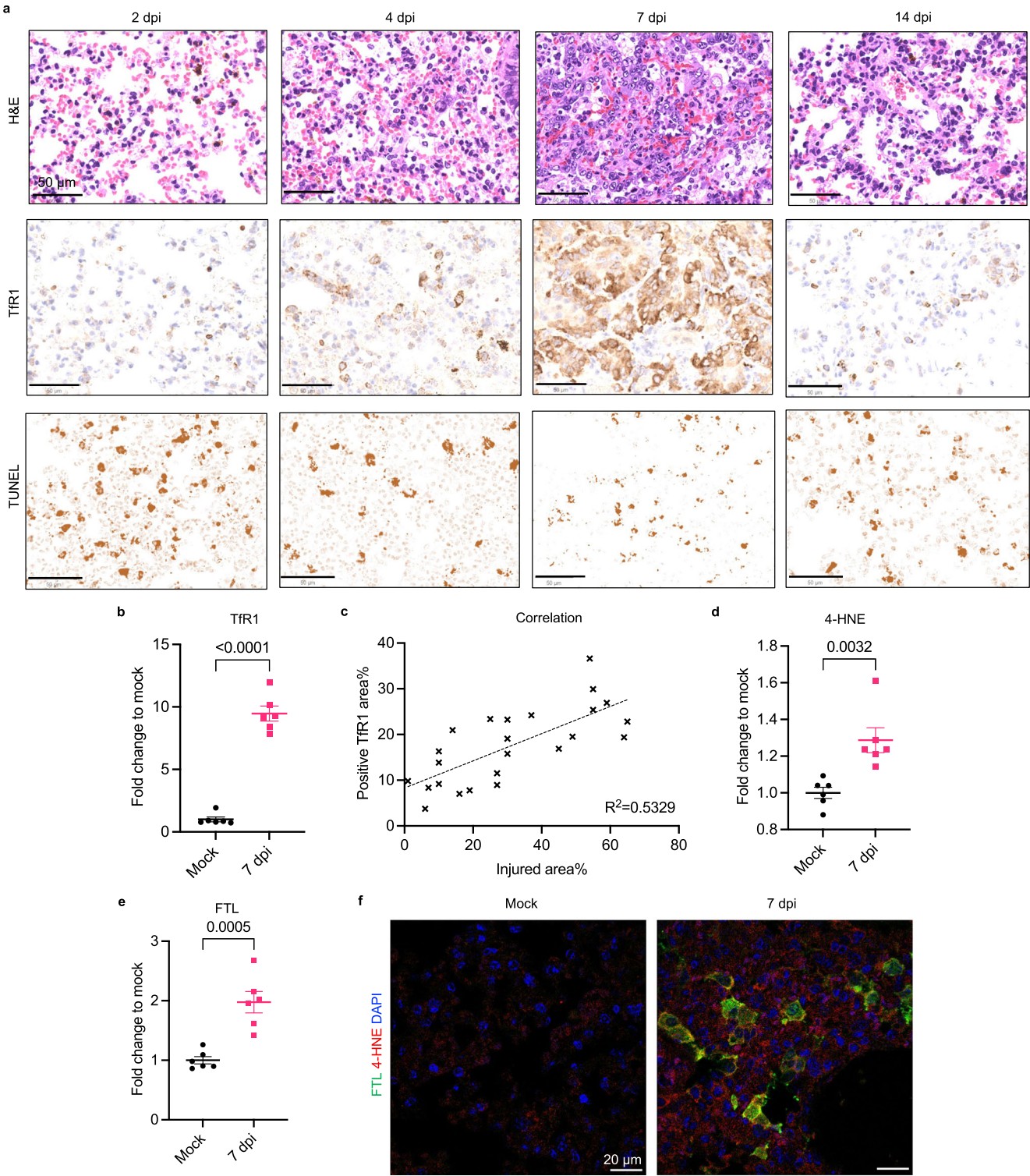

**Fig. 4 | Ferroptosis correlates with lung disease severity in a COVID-19 Syrian hamster model. a** 10-week-old male hamsters were inoculated with $10^5$ PFU SARS-CoV-2 and euthanized at several time points to examine lung tissue. Representative images showing H&E staining, IHC staining using anti-TfR1 antibody (clone H68.4), and TUNEL staining on SARS-CoV-2-infected hamster lung sections collected on 2, 4, 7, 14 days post infection (dpi). Scale bar = 50 μm. **b** The positive TfR1 stain area is normalized to the mock group. Data shown as mean ± SEM, $n = 6$ (mock), $n = 7$ (7 dpi), unpaired two-sided $t$ test ($p$ value indicated). **c** Injured area% and positive TfR1 area% in infected lungs ($n = 24$) were plotted and fitted with linear regression. $R^2 = 0.5329$. Mock and infected lung section were IF stained with anti-4-HNE antibody (clone HNEJ-1) and anti-FTL antibody. The mean intensities of (**d**), HNEJ-1 and (**e**), FTL were normalized to the mock group. Data shown as mean ± SEM, $n = 6$ for both groups, unpaired two-sided $t$ test ($p$ value indicated). **f** Representative images of FTL and 4-HNE stain on mock and infected hamster lung at 7 dpi. Nucleus is in blue, 4-HNE in red, and FTL in green. Scale bar = 20 μm.

suppressed by ferroptosis inhibitors, ferrostatin-1 and liproxstatin-1 (Fig. 2e). Untreated primary lung epithelial cells are more resistant to ferroptosis induced by GPX4 inhibition using RSL3, system xc⁻ (SLC7A11) inhibition using IKE, or FAC treatment alone, in comparison with a lung epithelial cancer cell line, Calu-1 (Fig. 2f, Supplementary Fig. 2f). This may be due to different dependencies on ferroptosis-regulatory pathways, including the GPX4-centered lipid repair pathway, among these two types of cells. However, FAC co-treatment greatly sensitized primary lung epithelial cells to RSL3 and IKE through the induction of lipid peroxidation. Our data suggest an important role of iron overload in promoting ferroptosis in lung cells during SARS-CoV-2 infection.

Multi-omics profiling and advanced bioinformatic analysis can facilitate the understanding of the molecular basis of pulmonary diseases. Single-nucleus transcriptomic profiling of COVID-19 lungs revealed important pathological hallmarks associated with SARS-CoV-2 infection including hyperinflammation, macrophage infiltration, alveolar epithelial cell exhaustion, parenchymal lung senescence, and fibrosis[27,39]. A Gene Ontology (GO) and Kyoto Encyclopedia of Genes and Genomes (KEGG) pathway enrichment analysis of published RNA-sequencing dataset revealed ferroptosis-related differentially expressed genes and important biological processes including the oxidative stress response, MAPK, and PI3K-AKT signaling pathway associated with COVID-19[40]. A weighted gene co-expression network analysis (WGCNA) of RNA-sequencing dataset revealed four key genes, PROK2, IL6, TNF, SLC7A11, closely related to COVID-19 ALI[41]. Our analysis of RNA-sequencing dataset (GEO, GSE171524) is consistent with these previous reports, emphasizing the ferroptosis pathway, particularly the iron regulatory pathway associated with severe COVID-19 lungs. Ferroptosis response is heterogeneous among different alveolar cell populations. This will facilitate future studies on designing targeted therapy to prevent ferroptosis in specific cell types.

Utilizing untargeted lipidomics analysis, we found significant depletion of phospholipids containing polyunsaturated fatty acyl tails (PL-PUFAs) and accumulation of lysophospholipids in severe COVID-19 lungs. These changes reflect the removal of oxidized PUFAs from the sn2 position of phospholipids and is a signature of ferroptosis[23]. The peroxidation of PL-PUFAs at bis-allylic carbons is catalyzed by ferrous iron and lipoxygenases through Fenton reactions. Phospholipase-A2-mediated cleavage of phospholipids at the sn2 position is a lipid repair mechanism and is upregulated during ferroptosis to detoxify lipid hydroperoxides. Moreover, lysophosphatidylcholine accumulation in the lung may increase the permeability of alveolar epithelium, disrupt lung surfactant, and stimulate pro-inflammatory lung microenvironment[42]. In contrast, lipid markers of apoptosis and necroptosis, including cardiolipins, ceramides, and phosphatidylinositols, were not elevated in COVID-19 lungs, which was consistent with our immunostaining results[43,44].

Pulmonary surfactant comprises a complex mixture of lipids and proteins that line the inner surface of the lung to prevent alveolar collapse and pathogen invasion[45]. Plasmalogens are the most abundant antioxidants in pulmonary surfactant, which protects against iron-induced lipid peroxidation[46]. We detected a reduction in numerous phosphatidylethanolamine (PE) plasmalogens containing vinyl-ether bonds in COVID-19 lungs (Supplementary Data 2). The depletion of plasmalogens reflects increasing oxidative stress in damaged lungs. We also found significant depletion in other surfactant lipids, dipalmitoyl-phosphatidylcholine, and palmitoyl-oleoyl-PG, in COVID-19 lungs (Supplementary Fig. 3a). DPPC is the most abundant phospholipid that maintains the pulmonary surfactant structure; phosphatidylglycerols, including palmitoyl-oleoyl-PG, playing a critical role in interacting with surfactant protein B (SP-B) in the alveolar surfactant layer[47]. Hydrolysis of palmitoyl-oleoyl-PG has been reported during surfactant injury in ARDS patient lungs[48]. Lung surfactant deficiency is associated with hyaline membrane disease, the pathological correlate of respiratory distress syndrome in infants[49]. Surfactant components and embolic lipid droplets have been reported in the hyaline membrane deposition in COVID-19 lungs[1,50]. Thus, depletion of surfactant phospholipids and accumulation of fatty acids, as seen in our lipidomics results, may contribute to the formation of hyaline membrane in COVID-19 ALI.

Ferroptosis contributes to severe lung tissue damage, as seen in both COVID-19 human lung samples and the hamster disease model. Through a longitudinal analysis of lung disease progression using Syrian hamsters, we found a strong correlation of ferroptosis markers—TfR1 and the lipid peroxidation product 4-HNE—with lung injury severity. Ferroptosis occurred at an early stage of lung disease and accompanied disease progression with high spatial correlation at lung lesion sites. Ferroptosis has been shown to drive inflammation-related diseases, such as ischemia-reperfusion injury; disease-associated molecular patterns (DAMPs) released by ferroptotic cells may initiate inflammation[51]. Elevated ferroptosis in the lung may complicate immune response and exacerbate tissue damage. We found that ferroptosis inhibitors, a ferrostatin-1 analog and liproxstatin-1, partially reduced lung injury and drug efficacy was limited by lung accumulation in the hamster model. These data suggest a driving force of ferroptosis during the progression of severe lung pathology and targeting ferroptosis in the lung as an alternative treatment. Pulmonary delivery of such treatments, such as inhalation, may be advantageous by directly targeting lung lesions and providing high bioavailability[52]. Antibody or ligand conjugation to target specific pulmonary cell types, such as endothelial cells or monocytes may also improve delivery capabilities[53]. The development of effective lung-targeting ferroptosis inhibitors is critical for effective ferroptosis suppression in the lung. Moreover, while we examined only SARS-CoV-2-infected samples, these findings may apply to other lethal coronaviruses, such as SARS and MERS.

In summary, we found that ferroptosis is associated with COVID-19 lung pathology. Accumulation of TfR1 and lipid peroxidation end products reflect the activation of ferroptosis in damaged tissues. Remarkable lipid changes also reflect compromised lung functions, oxidative stress, and lipid peroxidation. Iron overload promotes ferroptosis in infected lungs. Blocking ferroptosis thus represents an adjuvant approach to preserving lung function in COVID-19 patients. In addition, ferroptosis protein and lipid markers may be useful for diagnosis of COVID-19 disease severity in patients.

## Methods
### Experimental model and study participant details
**Human subject.** The study involving human subject was conducted under the Institutional Review Board of Columbia University (AAAT0388, approved 04/28/2020). All human samples were obtained from the Columbia University tissue bank. Severe COVID-19 lung autopsies with ALI and non-ALI pathology were collected from patients deceased from respiratory failure caused by SARS-CoV-2 infection. Severe lung explants were collected from patients with severe COVID-19 disease. Mild COVID-19 lung explants/biopsies were selected from patients recovered from SARS-CoV-2 infection. Control lungs were selected from resections of pneumothorax lungs and neoplastic lungs that had no lung injury and were not infected with SARS-CoV-2. Non-COVID-19 control lung autopsies with ALI were collected from patients deceased from respiratory failure before the pandemic. Both male and female patients were equally involved. The sex and age of all subjects involved are provided in Supplementary Data 1. Consents were obtained for all patient samples obtained from the Columbia University tissue bank. The initial and last laboratory ferritin test records were obtained from severe COVID-19 patients involved in this study. Patient serum was collected during routine blood test and an

immunoassay against ferritin was performed on the Roche analyzer in the Automated Lab.

**Animal model.** All animal studies were performed in compliance with the Guide for the Care and Use of Laboratory Animals of the National Institutes of Health and the Animal Welfare Act Code of Federal Regulations. Animal study involving SARS-CoV-2 infection in Syrian hamsters was approved by the Institutional Animal Care and Use Committee of Columbia University (AC-AABT6656, approved 12/15/2022). Animal work involving SARS-CoV-2 was conducted under appropriate biosafety precautions in a BSL-3 facility at the Herbert Irving Comprehensive Cancer Center, New York. 10-week-old male Golden Syrian hamsters (*Mesocricetus auratus*; outbred hamster strain 049, Charles River Labs) were housed in individual cages and had ad libitum access to food and water. Animals were allowed at least four days of acclimation before the start of study.

**Virus strain.** The SARS-CoV-2-USA-WA1/2020 (WA-1) strain was propagated and quantified in the BSL-3 facility at the Aaron Diamond AIDS Research Center, New York. The virus isolate was passaged in Vero E6 cells for 2 passages and the supernatant was harvested at 70 h post infection. The resultant supernatant was titrated in Vero E6 cells to obtain the TCID/mL value which was used to calculate the PFU value for animal experiments.

**Cell lines.** All mammalian cells were obtained from Columbia University Cancer Cell Bank and cultured following ATCC recommended protocols and in medium supplemented with 10% fetal bovine serum (FBS, Thermo Fisher Scientific 26400044) and 1% Penicillin-Streptomycin (Thermo Fisher Scientific 15140122). HT-1080 cells were cultured in Dulbecco's modified Eagle's medium (Corning-FISHER 10-013-CM) supplemented with 1% non-essential amino acids (Thermo Fisher Scientific 1140076). HT-29 and Calu-1 cells were cultured in McCoy's 5 A medium (Thermo Fisher Scientific 16600108). Primary human bronchial/tracheal epithelial cells (ATCC PCS300010) were cultured in airway epithelial cell basal medium (ATCC PCS300030) supplemented with airway epithelial cell supplement (ATCC PCS300040) and 0.1% Penicillin-Streptomycin. All cell lines were authenticated through STR profiling by the Cell Bank or ATCC.

## Method details
**Chemicals.** RSL3, IKE were synthesized by the Stockwell lab[23,29]. All other chemicals were purchased from qualified vendors: staurosporine, Selleck Chemicals S1421; Z-VAD-FMK, Abcam ab120382; LCL-161, ChemieTek CT-LCL161; TNFα, PeproTech 300-01 A; ferric ammonium citrate, Sigma-Aldrich F5879; hemoglobin, Sigma-Aldrich H7379; ferrostatin-1, Sigma-Aldrich SML0583; Liproxstatin-1, Selleck Chemicals S7699; necrostatin-1s, Cell Signaling Technology 17802 S. TH-2-31 ($N^2$,$N^3$-dicyclohexyl-5-(3-methyl-1,2,4-oxadiazol-5-yl)pyridine-2,3-diamine) was synthesized by Curia. The structure of Th-2-31 was confirmed by NMR: ¹H NMR (400 MHz, Chloroform-d) δ 7.81 (d, J = 1.7 Hz, 1H), 6.86 (d, J = 1.9 Hz, 1H), 4.04–3.78 (m, 1H), 3.19 (td, J = 10.0, 4.2 Hz, 1H), 2.44 (s, 3H), 2.01 (d, J = 10.6 Hz, 4H), 1.89 – 1.64 (m, 4H), 1.58 (d, J = 13.0 Hz, 1H), 1.47 – 1.01 (m, 9H).

**Antibodies.** The following antibodies and validation were used: TfR1 antibody (clone 3F3-FMA), originated from the Stockwell lab[18], validated in this study using human DLBCL xenograft tissue; MDA adduct antibody (clone 1F83), originated from the Uchida lab and validated in this study using human DLBCL xenograft tissue[19]; phospho-MLKL antibody (phospho S358, clone EPR9514), Abcam ab187091, RRID:AB_2619685; cleaved Caspase 3 antibody (Asp175), Cell Signaling Technology 9661 L, RRID:AB_2341188; TfR1 antibody (clone H68.4), Santa Cruz sc-65882, RRID:AB_1120670; cleaved Gasdermin D antibody (clone E7H9G), Cell Signaling Technology 36425, RRID:AB_2799099;

4-HNE antibody (clone HNEJ-1), originated from the Toyokuni lab[36]; p53 antibody (clone DO-1), Santa Cruz sc-126, RRID:AB_628082; MDM2 antibody (clone D1V2Z), Cell Signaling Technology 86934, RRID:AB_2784534; FTL antibody, Proteintech 10727-1-AP, RRID:AB_2278673; CD11c antibody (clone N418) conjugated with Alexa Fluor 488, Thermo Fisher Scientific 53-0114-82, RRID:AB_469903; CD68 antibody (clone FA-11), Abcam ab53444, RRID:AB_869007; α-Tubulin antibody (clone DM1A), Santa Cruz sc-32293, RRID: AB_628412; FSP1 antibody, Proteintech 20886-1-AP, RRID:AB_2878756; GPX4 antibody, Abcam ab125066, RRID:AB_10973901; goat anti-Mouse IgG (H + L) antibody Alexa Fluor 594, Thermo Fisher Scientific A-11032, RRID: AB_2534091; goat anti-Rabbit IgG (H + L) antibody Alexa Fluor 594, Thermo Fisher Scientific A-32740, RRID:AB_2762824; goat anti-Rat IgG (H + L) antibody Alexa Fluor 488, Cell Signaling Technology 4416, RRID:AB_10693769. All antibodies originated from research labs were previously validated in published studies and all commercially available antibodies were validated by manufacturers.

**Animal study.** For all animal studies, 10-week-old male Golden Syrian hamsters (*Mesocricetus auratus*; outbred hamster strain 049, Charles River Labs) were used. Hamsters were embedded with a temperature monitoring microchip in the nape and were randomized into different treatment groups. SARS-CoV-2 infected and mock infected animals were anesthetized with a mixture of 200 mg/kg ketamine and 10 mg/kg xylazine 1 h before intranasal inoculation procedure. The virus infected animals were intranasally inoculated with 0.1 mL PBS containing $10^5$ PFU SARS-CoV-2. The mock-infected animals were challenged with equivalent volume of PBS. To monitor lung disease progression, 9 hamsters were used and euthanized on day 2, day 4, day 7, and day 14 post-infection. Mock-infected animals were euthanized on day 14. For the Th-2-31 efficacy study, 18 hamsters were used and randomized into each group. SARS-CoV-2 and mock-infected hamsters were intraperitoneally injected with 20 mg/kg Th-2-31 or vehicle (3.3% DMSO v/v, 13% 2-hydroxypropyl β cyclodextrin w/v, 6.5% ethanol v/v, 30% PEG-400 v/v, 5% Tween80 v/v in water) 1 h after the viral challenge and repeatedly administered twice daily. All animals were euthanized 1 h after the first drug administration on day 7 post infection. Lung and plasma from cardiac puncture were collected from each animal. For the liproxstatin-1 efficacy study, 18 animals were used and randomized into each group. SARS-CoV-2 and mock infected hamsters were intraperitoneal injected with 8 mg/kg liproxstatin-1 or vehicle (2.5% DMSO v/v, 10% 2-Hydroxypropyl β cyclodextrin w/v in water) 1 h before the viral challenge and repeatedly administered once daily. All animals were euthanized 1 h after drug administration on day 6 post infection. Lung, heart, kidney, and plasma from cardiac puncture were collected from each animal. In addition, 6 dpi lung sections were prepared in the L.M. lab and harvested from Syrian hamsters inoculated with the same SARS-CoV-2 virus strain.

**Tissue pathology examination.** Lung tissue block smaller than 4 mm × 4 mm × 4 mm were fixed in 4% paraformaldehyde (PFA) for over 24 h. Tissues were washed three times with PBS and then paraffinized using Sakura VIP 5 tissue processor. Paraffinized tissues were sectioned at 5 μm and mounted onto slides. FFPE tissue sections were stained with hematoxylin and eosin (H&E) and scanned using Leica AT2 whole slide digital imaging at 40 x. The scanned whole slide images were annotated for injury and image border using QuPath (v0.4.3). Percentage injured area was calculated by dividing the sum of injured areas by total area and multiplying by 100%. For kidney and heart tissue evaluation, a practicing renal/cardiovascular pathologist blind to the animals' conditions/treatments performed light microscopic evaluated of H&E FFPE tissue from the sampled kidneys and hearts.

**Cultured cell immunofluorescence.** Cells were seeded at 0.1 million cells per well in 24-well plates containing 12 mm diameter circular

cover glasses 24 h before the experiment. HT-1080 cells were treated with 2 μM staurosporine for 3 h to induce apoptosis. HT-29 cells were treated first with 20 μM Z-VAD-FMK for 30 min and then treated with an addition of 10 μM Smac mimetic LCL-161 and 20 ng/mL human TNFα for 24 h to induce necroptosis. Negative control cells were treated with DMSO. Cells were then washed twice with PBS and fixed in 4% PFA (Fisher Scientific 50-980-494) in PBS containing 0.1% Triton X-100 (PBST, Sigma-Aldrich T9284) for 20 min at room temperature at dark. Cells were washed three times with PBST and blocked with 5% normal goat serum (NGS, Thermo Scientific 50062Z) at room temperature for 1 h. After blocking, cells were incubated with primary antibody diluted in 5% NGS overnight at 4 °C. Cells were washed with PBST three times and incubated with secondary antibody solution at room temperature for 1 h. Cells were washed three times with PBST. Cover glasses containing cells were added with ProLong diamond antifade mount with DAPI (Thermo Scientific P36962) and mounted onto slides. Slides were air dried for at least 24 h before imaging.

**Tissue immunofluorescence.** FFPE slides were deparaffinized in three changes of xylene (Sigma-Aldrich 534056) and rehydrated in two exchanges of 100%, 95%, and 75% ethanol for 5 min each. Antigen retrieval was performed by incubating slides in Tris-EDTA Buffer (10 mM Tris Base, 1 mM EDTA Solution, 0.05% Tween 20, pH 9.0) for 10 min at 95 °C. Slides were cooled to room temperature and rinsed twice with deionized water. Sections were permeabilized by incubating in two exchanges of PBS containing 0.4% Triton X-100 each for 10 min. Sections were then blocked with 10% normal goat serum for 1 h at room temperature. Sections were incubated with primary antibody overnight at 4 °C. The following dilutions of primary antibody were used: TfR1 antibody (clone 3F3-FMA), 1:250; MDA adduct antibody (clone 1F83), 1:1,000; phospho-MLKL antibody (phospho S358, clone EPR9514), 1:100; cleaved Caspase 3 antibody (Asp175), 1:500; 4-HNE antibody (clone HNEJ-1), 1:100; p53 antibody (clone D0-1), 1:500; MDM2 antibody (clone D1V2Z), 1:500; FTL antibody, 1:250; CD11c antibody (clone N418) conjugated with Alexa Fluor 488, 1:100; CD68 antibody (clone FA-11), 1:100; TfR1 antibody (clone H68.4), 1:1000 for IHC staining. Sections were washed three times with PBST each for 10 min and then incubated with secondary antibody for 1 h at room temperature. Sections were washed three times with PBST each for 10 min. Sections were applied with VECTASHIELD antifade mounting medium with DAPI (Vector Laboratories H-2000) and covered with clean cover glasses.

**Image collection and analysis.** All immunofluorescent slides were imaged by Zeiss LSM800 confocal microscope using 63 x or 40 x Oil DIC objective using Zeiss Zen Blue (v2.1) software. All imaging parameters including laser intensity and scan area remained constant for each experiment. Images were captured randomly across the tissue section to ensure unbiased data collection. Quantification of the fluorescence intensity in each cell region including the nucleus and the cytoplasm was performed using CellProfiler (v4.0.7, CellProfiler Image Analysis Software, RRID:SCR_007358)[54] and QuPath (v0.4.3, Quantitative Pathology & Bioimage Analysis, RRID:SCR_018257)[55]. Nuclei were first identified using a cell detection tool based on the intensity of the DAPI channel and the cytoplasm boundary was then identified by cell expansion around the nuclei. The fluorescence intensity of secondary antibody in the cytoplasm above the threshold was measured. Immunohistochemistry slides were imaged using Leica AT2 Auto bright field whole slide scanning system at 40 x. Whole slide images were quantified on QuPath (v0.4.3) using thresholding method to identify positive staining area.

**Th-2-31 plasma stability test.** Hamsters were randomized into 3 injection groups and treated with a single dose of Th-2-31 at 10 mg/kg through intravenous injection ($n = 6$) or at 20 mg/kg through intraperitoneal injection ($n = 6$) or oral administration ($n = 6$). Hamsters were bled 150 μL blood through retro-orbital vein under isoflurane anesthesia at 0, 0.5, 1, 2, 4, and 8 h after single dosing, with $n = 3$ hamsters per timepoint and each individual hamster covering 3 timepoints. In each treatment group, lungs were collected from hamsters euthanized at 2 h or 24 h. Lungs and plasma were flash-frozen in liquid nitrogen and stored at −80 °C prior to extraction.

**Compound extraction from tissue and plasma.** For Th-2-31, extraction from plasma was performed by adding 900 μL acetonitrile to 100 μL plasma, sonicating to mix, and incubating on ice for 2 h for optimal extraction. For tissue extraction, approximately 100 mg tissue was weighed and placed in the bead homogenizer tube (Omni 19-628). MS grade water was added to samples to make 500 mg/mL homogenate. Samples were homogenized for 4 cycles of 30 s at highest speed on the shaker. 100 μL tissue homogenate was mixed with 900 μL acetonitrile, sonicated, and incubated on ice for 2 h. Both plasma and tissue extracts were centrifuged at $4000 \times g$ at 4 °C for 10 min, and the supernatant was collected and dried using GeneVac. Dried samples were redissolved in acetonitrile:water (1:1; v/v) and analyzed on LC-MS. For Liproxstatin-1, extraction from plasma was performed by adding 250 μL methanol:acetonitrile:water (4:4:1; v/v/v) to 50 μL plasma. Samples were incubated for 1 h and centrifuged at $14,000 \times g$ at 4 °C for 20 min. The supernatant was analyzed on LC-MS.

**Untargeted lipidomics**

**Sample preparation.** COVID-19 lung samples procured during routine gross examination in surgical pathology were deposited into 2 mL cryovials, snap-frozen in liquid nitrogen, filled with OCT, and then stored at −80 °C. OCT-embedded blocks were then sectioned at 10 μm thickness and washed 5 times with PBS as described previously to remove OCT[56]. Briefly, 3 mL of ice-cold PBS was added to each vial containing 20 tissue sections, vortex-mixed, and centrifuged at $2000 \times g$ at 4 °C for 10 min. Following centrifugation, the supernatant was aspirated carefully to avoid taking up any tissue. This cycle of washing the tissues with cold PBS combined centrifugation was repeated 4 times. The lung sections were then incubated in methanol/acetonitrile (1:1; v/v) at −80 °C for 24 h to inactivate SARS-CoV-2 virus[57]. Viral deactivation solvents were discarded. Lipids were extracted using a modified Folch extraction method enabling a multi-omics analysis[58]. Briefly, the tissues were homogenized in 300 μL of methanol containing 0.01% BHT and a mixture of lipid standards using a micro-tip sonicator. Then 600 μL of dichloromethane was added to homogenized samples, and vortex-mixed for 30 s followed by incubating on dry ice on a shaker for 2 h. Next, 180 μL of ice-cold water was added to each sample, vortex-mixed, and incubated for 30 min on ice before centrifugation at 1500 x g for 20 min at 4 °C. The total lipid extract was transferred into a glass vial followed by drying under a gentle stream of nitrogen gas. The dried samples were reconstituted in a solution of IPA/ACN/water (4:3:1; v/v/v) before lipidomics analysis. The protein pellet was removed to quantify proteins using the BCA assay.

**LC-MS data acquisition.** Lipids were analyzed by LC-MS as previously described[59–61]. In brief, samples were analyzed using Acquity UPLC I-class PLUS interfaced with a Synapt G2-Si Mass spectrometer (Waters Corp.). Chromatographic separation was performed with a 20 min gradient elution profile on a Waters Acquity CSH C18 column (1.7 μm, 2.1 mm×100 mm). Both positive and negative electrospray ionization (ESI) modes used the same mobile phase: (A) water:ACN (40:60; v/v) and (B) water:ACN:IPA (5:10:85; v/v/v). However, different mobile phase modifiers were used in order to improve the lipid coverage[62]. For the positive mode, 10 mM ammonium formate and 0.1% formic acid were used, while 10 mM ammonium acetate was used for the negative ESI mode. The following linear gradient at 400 μL/min flow rate with a column temperature at 55 °C was used: 0−2 min: 60%B, 2−2.3 min: 75%

B, 2.3–10 min: 90%B, 10–17 min: 100%B, 17–20 min: 40%B. The Synapt G2-Si mass spectrometer equipped with a LockSpray ion source and was operated in both ESI. modes over the mass range of 50–1600 m/z. Source voltages were set to ± 2.5 kV, 30 V, and 5 V for capillary, sampling, and extraction cones, respectively. Moreover, the temperature was set to 120 °C for the source and 500 °C for sample desolvation. Gas flow rates were set at 900 L/h and 50 L/h for the desolvation gas and cone gas, respectively. Fragment ion spectra were generated using enhanced data-independent ion mobility (HDMSE) acquisition mode where data from mobility separation ions are collected in two channels with either low collision energy applied at 4 V or with an elevated collision energy ramp from 25 to 60 V for precursor and fragment ions, respectively. Nitrogen as the drift gas was held at a flow rate of 90 mL/min in the IMS cell with a wave velocity of 600 m/s and a wave height of 40 V.

**LC-MS data analysis and annotations.** The raw LC-MS data files were converted to the netCDF format using the DataBridge tool implemented in MassLynx software (v4.2, Waters). Then, they were subjected to peak-picking, retention time alignment, and grouping using the XCMS package in the R (v4.0) environment. Statistical analysis was performed in R (v4.0) environment and MetaboAnalyst R (v3.0). The extracted lipid features were normalized by total protein concentrations, median normalization, and log-transformed within each sample before statistical analyses. Group differences were calculated using the Welch t-test. The Benjamin–Hochberg (BH) method was also applied for multiple testing corrections, and the adjusted $p$ values were calculated to control the false discovery rate for each test. The adjusted $p$ values lower than 0.05 were considered statistically significant (FDR-corrected p < 0.05) with a fold change threshed of 1.5 was considered as statistically significant. Identification of detected lipid features was initially obtained by searching monoisotopic masses against the available online databases such as METLIN, Lipid MAPS, and HMDB with a mass tolerance of 5 ppm. Moreover, isotopic patterns and retention time of each lipid precursor ion were assessed. Structural elucidation of lipids confirmed by inspecting the diagnostic fragment ions and corresponding acyl chain fragment information obtained by UPLC-HDMSE. The nomenclature and shortened annotation of lipids proposed by the Lipid Maps Consortium were employed[63].

**Cell viability assay.** Cells were seeded in 384-well format culture plate (Thermo Fisher Scientific 164610) at 1000 cells per well and incubated overnight. Cells were treated with conditions specified in figure legends. Cell viability was measured by CellTiter-Glo (Promega G7573) using a luminescent plate reader according to the manufacturer's instructions. For all viability assays, at least 2 biological replications were performed.

**Lipid peroxide measurement.** Cells were seeded in 6-well plates at 0.5 million cells per well and incubated overnight. Cells were treated with conditions specified in figure legends and washed once with HBSS (Thermo Fisher Scientific 14025092). Cells were then incubated with HBSS containing 2 μM C11-BODIPY[581/591] (Invitrogen D-3861) at 37 °C for 30 min in dark. Cells were trypsinized (Invitrogen 25200-114), washed once with HBSS, and resuspended in fresh HBSS for flow cytometry analysis. Samples were analyzed using Beckman CytoFLEX System B4-R0-V0 flow cytometer and excitation at 488 nm and emission at FITC channel with 525/40 nm bandpass filter for oxidized C11-BODIPY. 10,000 events after gating were collected for each sample. Flow cytometry data were processed in FlowJo (v10.8). Cells were gated for live cells in SSC-A vs. FSC-A plot and for singlet cells in FSC-H vs. FSC-A plot. The FITC-A histogram of singlet cells was plotted. For all flow cytometry assays, at least 2 biological replications were performed.

**Western blot.** Approximately 3 million cells were collected and washed twice with PBS. Cell pellet was lysed in 40 μL RIPA buffer (Thermo Fisher Scientific 89901) containing protease inhibitor cocktail (Sigma-Aldrich 11697498001) and incubated on ice for 10 min. Lysate was centrifuged at 17,000 × $g$ for 15 min 4 °C. The supernatant was collected, and protein amount in cell lysate was quantified using Pierce™ BCA Protein Assay Kit (Thermo Fisher Scientific 23225) following the manufacturer's protocol. In brief, the standard curve was generated by serial dilutions of BSA standards. Both standards and samples were added in 96-well format clear assay plate (Greiner Bio-One 655101) in duplicates and incubated with the colorimetric working solution at 37 °C for 30 min at dark. The plate was measured using the plate reader and detected by absorbance at 562 nm. The concentration of sample is interpolated from the standard curve. Lysate was diluted with 3x blue loading buffer containing 125 μM DTT (Cell Signaling Technology 7722 S) and incubated at 80 °C for 10 min. An equal amount of protein in the range of 30–50 μg was loaded in each lane of the NuPAGE 4–12% Bis-tris gel (Fisher Scientific WG1401) and transferred onto nitrocellulose membranes (Thermo Fisher Scientific IB23002) using electrophoretic semi-dry western blot transfer system. Membranes were blocked with TBS blocking buffer (Thermo Fisher Scientific 37535) for 1 h at room temperature and incubated with primary antibody diluted in TBS with 0.1% Tween20 (TBS-T): blocking buffer 1:1 overnight at 4 °C. The following dilutions of antibody were used: α-Tubulin antibody (clone DM1A), 1:10,000; FSP1 antibody, 1:500; GPX4 antibody, 1:250; FTL antibody, 1:500. Membranes were washed three times in TBS-T and then incubated with secondary antibodies diluted at 1:10,000 in 1:1 TBS-T: blocking buffer for 1 h at room temperature. Membrane was washed three times in TBS-T and imaged on LI-COR Odyssey infrared imaging system. Images were collected and optimized contrast settings using the Image Studio software.

**Analyzing RNA-sequencing data.** Reanalysis of published dataset was performed on Broad Institute Single Cell Portal. UMAP scatter plot was performed by separating the COVID-19 and control samples for each gene of interest. Dot plot was performed to compare the expression level of each gene between the COVID-19 and the control group.

### Statistical analysis
Statistical analysis including unpaired $t$ test and one-way ANOVA was performed using GraphPad Prism 9 in most figures unless otherwise stated in the figure legend. Data are plotted as mean ± SEM or mean ± SD as stated in the figure legend. No data or samples were excluded from the study.

### Reporting summary
Further information on research design is available in the Nature Portfolio Reporting Summary linked to this article.

## Data availability
The mass spectrometry data generated in this study have been deposited in the EMBL-EBI MetaboLights database under accession code MTBLS3107. The rest data generated in this study are provided in the Supplementary Information/Source Data file. The single cell RNA-sequencing data used in this study are available in the GEO database under accession code GSE171524 [https://doi.org/10.1038/s41586-021-03569-1]. Source data are provided with this paper.

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

## Acknowledgements

S.T. is supported in part by funding JST CREST (JPMJCR19H4). This research was funded in part through the NIH/NCI Cancer Center Support Grant P30CA013696 and utilized the Molecular Pathology Shared Resource for all tissue processing and immunohistochemical staining as well as its Tissue Bank for providing COVID-19 and control lung samples. We thank Hongyan Tang for performing the immunohistochemistry staining. We thank Dr. Michael L. Miller for verifying the tissue pathology quantification method. We appreciate Dr. Manoj S. Nair for preparing the SARS-CoV-2 virus strain. We thank Dr. Carol Prives for providing the p53 antibody. We are grateful for hamster lung tissues provided by Dr. Adolfo García-Sastre, Soner Yildiz, Thomas Kehrer, and Anastasija Cupic during preliminary experiments.

## Author contributions

B.Q. performed all immunofluorescent staining experiments and analysis, cell line experiments, and design of animal study. F.Z. performed all mass spectrometry-based lipidomics analysis on COVID-19 patient lung autopsies and compound quantification in animal samples. A.S. performed histopathological examination and quantification of COVID-19 patient lung samples and Syrian hamster lung sections. C.C. performed surgical operations in Syrian hamsters, including SARS-CoV-2 inoculation, compound administration, and tissue harvesting. M.S. performed histopathological examination on Syrian hamster heart and kidney sections. H.T. performed development and characterization of ferrostatin-1 analog Th-2-31. Antibody reagents and tissue samples were provided by K.U., S.T., L.M., H.H. Experimental design and execution were overseen by B.R.S. Manuscript was drafted by B.Q. and F.Z. and reviewed and edited by B.R.S., B.Q., F.Z., A.S. S.T., H.H., L.M.

## Competing interests

B.R.S. is an inventor on patents and patent applications involving ferroptosis; co-founded and serves as a consultant to ProJenX, Inc. and Exarta Therapeutics; holds equity in Sonata Therapeutics; serves as a consultant to Weatherwax Biotechnologies Corporation and Akin Gump Strauss Hauer & Feld LLP. B.R.S. and H.T. are inventors on a patent application related to ferrostatin-1 analog development. B.Q. is now a Post Doc Scientist at Gilead Sciences, Inc. The remaining authors declare no competing interests.

## Ethics & Inclusion statement

This study included local researchers throughout the research process. The roles and responsibilities were agreed amongst all collaborators ahead of the research. This study did not result in stigmatization, incrimination, discrimination, or otherwise personal risk to participants. This research involved safety risk to researchers handling SARS-CoV-2 infected samples, but all procedures were approved by IRB and IACUC protocols, and the study was performed in biosafety cabinets to ensure no exposure to active viruses. Local and regional research relevant to this study were included in citations.

## Additional information

**Peer review information** : *Nature Communications* thanks Antonio Solimando and the other, anonymous, reviewer(s) for their contribution to the peer review of this work. A peer review file is available.

