## [Peer Review File · Nature Communications]

Fatal COVID-19 pulmonary disease involves ferroptosisReviewers' Comments:

Reviewer #1:

Remarks to the Author:

To the authors

The authors report that SARS-CoV-2 infection can cause severe lung diseases like pneumonia and acute respiratory distress syndrome, though the underlying mechanisms are not well understood, and effective treatments are limited. They found evidence that SARS-CoV-2 may cause ferroptosis, an iron-dependent cell death, in the lungs of COVID-19 patients. By analyzing lung tissue from fatal COVID-19 cases, the authors detected indicators of ferroptosis, including increased transferrin receptor 1, lipid peroxidation, and altered lipid metabolism. These ferroptosis markers were elevated in both acute lung injury and non-acute lung injury COVID-19 lungs. Ferroptosis is also strongly correlated with lung damage in a COVID-19 animal model. Overall, the results reveal ferroptosis as a critical process driving lung injury and disease in COVID-19. The authors suggest dietary and drug-based inhibition of ferroptosis as potential therapies to prevent pulmonary damage during SARS-CoV-2 infection. This study elucidates a pathological mechanism of COVID-19 lung diseases and proposes novel treatment strategies targeting ferroptosis.

Here are some potential limitations and suggestions for improving the methods:

Limitations:

Small sample size of human lung tissues, especially for the control group - additional control samples could strengthen conclusions.

Only examined lung tissue - changes in other organs unknown.

The mechanism linking ferroptosis to lung pathology not fully elucidated.

Single virus strain used in animal model - different strains should be tested.

Lipidomic analysis has potential confounders - careful interpretation is needed.

Moreover, The study only examined end-stage lung tissue from deceased COVID-19 patients.

Analyzing lung samples from patients at earlier stages of the disease could provide more insight into how ferroptosis develops over time.

Only one control group (pneumothorax patients) was used. Including additional control groups, like patients with other respiratory illnesses, would allow better differentiation of changes specific to COVID-19.

The timeline of ferroptosis onset during infection is unclear. Longitudinal sampling in animal models could determine if it is an early or late event.

How SARS-CoV-2 triggers ferroptosis is unknown. Investigating the interactions between viral proteins and ferroptosis pathways could reveal the mechanism.

The contribution of ferroptosis to lung pathology relative to other cell death pathways was not defined. Comparisons of different cell death markers could quantify which are most prominent.

It is unknown if inhibiting ferroptosis after symptom onset could still be beneficial. Testing ferroptosis inhibitors at different time points during infection would address this.

Lipidomic analysis showed associations, but causality is uncertain. Directly manipulating lipids shown to be altered could demonstrate their functional roles.

Effects of lipid changes on lung surfactant and pulmonary mechanics were not examined. Biophysical

studies on surfactant function would be informative.
Suggestions for improvement:

Increase sample size with age, sex, and comorbidity-matched controls.
Examine other relevant tissues like the heart, kidney etc.
Conduct additional experiments to clarify the relationship between ferroptosis and lung damage.
Test multiple clinically relevant SARS-CoV-2 strains in vivo.

Validate lipidomic findings using additional techniques like mass spectrometry imaging.
Assess ferroptosis markers and lipid changes at single cell level.
Evaluate other cell death pathways besides apoptosis, necroptosis and ferroptosis.
Determine if observed lipid changes impact lung surfactant function.
Optimize single-cell assays to measure ferroptosis response heterogeneity across lung cell populations.

Assess the biophysical properties of surfactant with lipid composition changes observed in COVID-19.

Overall, the study provides evidence that ferroptosis contributes to lung injury in COVID-19, but further mechanistic work and validation in additional models would strengthen the conclusions. The suggestions could help elucidate ferroptosis and lipid changes in more detail and assess the therapeutic potential of inhibiting this pathway. If beyond the scope of the manuscript, these limitations should be highlighted and discussed. Finally, this reviewer personally misses some insights and revision of the available literature regarding SARS-CoV-2, WGCNA, bronchial epithelium cells with explainable artificial intelligence and transcriptional profiling can guide novel treatment. The introduction and discussion sections would benefit from this standpoint. Indeed, The mechanistic studies on ferroptosis regulators and lipid enzymes could reveal how SARS-CoV-2 disrupts these pathways to cause lung epithelial cell death. This could uncover new genes and proteins involved in COVID-19 pathogenesis.

The animal model experiments testing ferroptosis inhibitors at different time points could help determine if these drugs have potential as COVID-19 treatments and when they would be most effective during illness.

Analyzing ferroptosis markers longitudinally in patient samples may identify biomarkers to predict who will develop more severe lung disease.

Examining the correlation between specific lipid changes, hub genes/proteins levels, and disease severity could highlight lipid species and genes contributing to pathogenesis.

The drug-target network analysis could be expanded by including ferroptosis regulators and lipid enzymes as potential targets. Drugs that modulate these could be candidates.

The single-cell assays could characterize if there is heterogeneity in the ferroptosis response across lung cell subpopulations, which may influence susceptibility.

Integrating the ferroptosis mechanisms with genomic, transcriptomic and lipidomic analyses could provide a more comprehensive picture of pathogenic pathways in COVID-19. The suggestions could yield new genes of interest, biomarkers, and targeted treatment strategies to improve patient outcomes.

Reviewer #2:
Remarks to the Author:

Qiu et al describe a ferroptosis signature in acute lung injury of COVID19 patients (n=11) as well as Syrian hamster COVID19 model.

Overall, experiments seem to be well performed using different experimental approaches.

Major concerns by this reviewer:

- One can question novelty considering previous reports describing ferroptosis in COVID-19 patients (Refs 18-23).
- A causative role for ferroptosis cannot be claimed with current data set. For this at least, a therapeutic setting using ferroptosis inhibitor (Lip1, UAMC-3203...) should be used in hamster model showing protection against injury.
- COVID19 patients are known to be very heterogenous in relation to pathology, comorbidities, medication/treatments... Therefore, a correlation to patient's clinical data and plasma injury markers is crucial.
- The observed changes in lipid composition fits with what we would expect after ferroptosis cell death wave. However, other biochemical/inflammatory processes can't be excluded. Therefore, adding an oxidative lipidomics analysis would strengthen the conclusion of the paper.

Reviewer #3:

Remarks to the Author:

This is an important manuscript that lays the foundation for ferroptosis as a primary driver of cell death in the lung tissue. This is a significant finding that provides a novel target to treat severe ARDS induced by COVID, which is still an unmet clinical need. They convincingly demonstrated ferroptosis in human lung tissue with COVID infections. They also demonstrated that markers of necroptosis and apoptosis were not elevated in the COVID lungs samples. And this was followed by LC-MS lipidomic analysis which showed a depletion of PUFA containing phospholipids and accumulation of lyso-phospholipids which is a signature of ferroptosis driven lipid peroxidation. They confirmed the findings from the human tissues in hamsters infected with SARS-CoV2. Overall, they provided a convincing case for the discovery of ferroptosis as a major driver of cell death in the lungs of SARS-CoV2 infected cases. All controls and methods were appropriate. I have a few comments and questions below.

The authors did not discuss the role ferroptosis regulators GPX4 or FSP1. It does not appear that GPX4 is expressed in lung tissue can authors comment on what is controlling ferroptosis in lung tissue? or have any evidence for GPX4 or FSP in lung.

Method question: The lipid extraction describes extraction from 10 µm sections was it a single section or multiple combined or could they provide an estimate of tissue amount?

To inactivate the virus, they incubated the tissue in methanol/acetonitrile mixture which can extract lipids, was this inactivation extract discarded or combined with the Folch extract?

Reviewer Comments

Reviewer #1 (Remarks to the Author):

To the authors

The authors report that SARS-CoV-2 infection can cause severe lung diseases like pneumonia and acute respiratory distress syndrome, though the underlying mechanisms are not well understood, and effective treatments are limited. They found evidence that SARS-CoV-2 may cause ferroptosis, an iron-dependent cell death, in the lungs of COVID-19 patients. By analyzing lung tissue from fatal COVID-19 cases, the authors detected indicators of ferroptosis, including increased transferrin receptor 1, lipid peroxidation, and altered lipid metabolism. These ferroptosis markers were elevated in both acute lung injury and non-acute lung injury COVID-19 lungs. Ferroptosis is also strongly correlated with lung damage in a COVID-19 animal model. Overall, the results reveal ferroptosis as a critical process driving lung injury and disease in COVID-19. The authors suggest dietary and drug-based inhibition of ferroptosis as potential therapies to prevent pulmonary damage during SARS-CoV-2 infection. This study elucidates a pathological mechanism of COVID-19 lung diseases and proposes novel treatment strategies targeting ferroptosis.

Here are some potential limitations and suggestions for improving the methods:

Limitations:

1. Small sample size of human lung tissues, especially for the control group - additional control samples could strengthen conclusions.

We thank you for this comment and suggestion. As suggested, we have expanded the sample size of both control cases (from 9 to 34) and COVID-19 cases (from 21 to 34). In total, we have the following groups in this study:

- 1) Control cases:
 - a. Negative control:
 - i. Normal lung tissue next to lung neoplasm (n=11)
 - ii. Normal lung tissue from pneumothorax patients (n=10)
 - b. Non-COVID lung with acute lung injury (n=13)
- 2) COVID-19 cases:
 - a. Severe autopsy from deceased COVID patient (n=21)
 - b. Severe explant from COVID patient (n=7)
 - c. Mild explant/biopsy from recovered COVID patient (n=6)

2. Only examined lung tissue - changes in other organs unknown.

We thank you for this comment and suggestion. We only examined the patient lung pathology because the lung is the most affected organ post SARS-CoV-2 infection. As suggested, we examined changes in other organs including the heart and kidney from our hamster study post SARS-CoV-2 infection. We did not observe any acute or chronic pathologies in other organs,

including the heart (cardiomyocytes, interstitium, and vasculature) and the kidney (glomeruli, tubules, interstitium, and vasculature). These data can be found in Supplementary Figure 2.

3. The mechanism linking ferroptosis to lung pathology not fully elucidated.

We thank you for this comment and suggestion. As suggested, we investigated the efficacy of ferroptosis inhibitor, liproxstatin-1 and ferrostatin-1 analog Th-2-31, in the hamster model. We found that repeated dosing of Th-2-31 and liproxstatin-1 for 7 and 6 days, respectively, alleviated lung injury compared to vehicle-treated animals. The ferroptosis marker, TfR1 showed a reduction in the inhibitor-treated group compared to the vehicle-treated group. These data suggest that ferroptosis contributes to lung pathology. These data can be found in the result section “Ferroptosis correlates with lung disease severity in a COVID-19 Syrian hamster model” and in Figure 4f, Extended Data Figure 4g-i.

4. Single virus strain used in animal model - different strains should be tested.

We thank you for this comment and suggestion. We used the wild type strain in the hamster study because it's the same strain found in COVID-19 patients used in this study. Therefore, the hamster model recapitulated the human lung disease and was used to study the disease mechanism.

5. Lipidomic analysis has potential confounders - careful interpretation is needed.

We thank you for this comment. We acquired COVID-19 samples from the same cohort of patients that were infected with SARS-CoV-2 (same timeframe, same strain, same hospital) to minimize differences among the same group.

6. Moreover, the study only examined end-stage lung tissue from deceased COVID-19 patients. Analyzing lung samples from patients at earlier stages of the disease could provide more insight into how ferroptosis develops over time.

We thank you for this comment and suggestion. As suggested, we analyzed human autopsy samples with mild pathology in comparison with severe pathology. The mild cases were lung explant/biopsy from patients who recovered from COVID-19 and the severe cases were COVID-19 lung explants with severe lung pathology. We found that severe cases had higher levels of the ferroptosis marker TfR1 and ferritin light chain compared to the mild cases, indicating that iron dysregulation is associated in severe cases and triggers ferroptosis. These data can be found in the results section “Dysregulation of iron homeostasis contributes to ferroptosis” and Figure 2b.

7. Only one control group (pneumothorax patients) was used. Including additional control groups, like patients with other respiratory illnesses, would allow better differentiation of changes specific to COVID-19.

We thank you for this comment and suggestion. As suggested, we have expanded the sample size of control cases (from 9 to 34). In total, we have the following control groups in this study:

- 1) Negative control:
 - a. Normal lung tissue next to lung neoplasm (n=11)

- b. Normal lung tissue from pneumothorax patients (n=10)
- 2) Non-COVID control:
 - a. Non-COVID lung with acute lung injury (n=13)

Interestingly, we found that the ferroptosis marker TfR1 and ferritin are specifically elevated in COVID-19 lungs but not non-COVID ALI lungs, suggesting that ferroptosis activation is specific to SARS-CoV-2 infection. These data can be found in the result section “Ferroptosis is elevated in post-mortem COVID-19 lungs” and Figure 1c.

8. The timeline of ferroptosis onset during infection is unclear. Longitudinal sampling in animal models could determine if it is an early or late event.

We thank you for this helpful comment and suggestion. We have included the longitudinal sampling of animal tissue at 2, 4, 7, and 14 days post infection. We found that lung pathology and ferroptosis markers, TfR1 and 4-HNE, progressed in the same pattern, increasing during the initial 7 days and finally resolved on day 14. Expression of TfR1 also correlates with the severity of lung injury. This suggest that ferroptosis onset is early and accompanies with disease progression. These data can be found in the result section “Ferroptosis correlates with lung disease severity in a COVID-19 Syrian hamster model” and Figure 4a-c, Extended Data Figure 4e.

9. How SARS-CoV-2 triggers ferroptosis is unknown. Investigating the interactions between viral proteins and ferroptosis pathways could reveal the mechanism.

We thank you for this important comment. As suggested, we investigated the ferroptosis pathway, particularly the iron metabolism pathway in patient samples, human lung cells, and the hamster disease model. We found upregulation of iron regulatory proteins including the transferrin receptor 1 (TfR1) and ferritin light chain (FTL) in patient lung samples. Besides, COVID-19 patients had hyperferritinemia. In addition, by analyzing a published single-cell RNA sequencing dataset [(GEO, GSE171524) of COVID-19 lung samples obtained from the same cohort of patients used in our study], we found upregulation of TfR1, FTL, and ferritin heavy chain 1 (FTH1) in COVID-19 lung samples, particularly overexpressed in alveolar macrophages, fibroblasts, and other epithelial cells. In addition, we generated an iron-overload model in human primary lung epithelial cells and the lung cancer cell line, Calu-1, using ferric ammonium citrate (FAC). FAC treatment induced lipid peroxidation accumulation in primary lung epithelial cells. FAC also induced ferroptotic cell death in ferroptosis-sensitive Calu-1 cells and synergized with ferroptosis inducers RSL3 and IKE in ferroptosis-resistant primary lung epithelial cells. Moreover, we found significantly increased FTL in hamster lungs at 7 days post SARS-CoV-2 infection compared with mock-infected animals.

Together, these data confirm that *iron overload* is associated with increased FTL expression and lipid peroxidation accumulation that contributes to ferroptosis in the lung during SARS-CoV-2 infection. These data can be found in the result section “Dysregulation of iron homeostasis contributes to ferroptosis” and Figure 2, Extended Data Figure 2.

10. The contribution of ferroptosis to lung pathology relative to other cell death pathways was not defined. Comparisons of different cell death markers could quantify which are most

prominent.

We thank you for this comment and suggestion. We have examined other cell death markers in patient samples, including cleaved caspase 3 for apoptosis, phospho-MLKL for necroptosis, and cleaved gasdermin D for pyroptosis. Among all the cell death marker tested, only ferroptosis markers, TfR1 and MDA, showed significant elevation in COVID-19 lungs, suggesting that ferroptosis is the most prominent cell death pathway activated in diseased lung tissue. These data can be found in the result section “Ferroptosis is elevated in post-mortem COVID-19 lungs” and Figure 1e, Extended Data Figure 1c.

11. It is unknown if inhibiting ferroptosis after symptom onset could still be beneficial. Testing ferroptosis inhibitors at different time points during infection would address this.

We thank you for this suggestion. As discussed in comment #3, we investigated the efficacy of ferroptosis inhibitor ferrostatin-1 analog Th-2-31 and liproxstatin-1 in the hamster model. We intraperitoneally injected Th-2-31 and liproxstatin-1 daily for 7 and 6 days, respectively, after one-time viral infection. We found that the drug-treated group showed ameliorated lung injury and lung ferroptosis compared to the vehicle-treated animals. These data suggest that inhibiting ferroptosis is beneficial in reducing lung pathology during infection. These data can be found in the results section “Ferroptosis correlates with lung disease severity in a COVID-19 Syrian hamster model” and in Figure 4f, Extended Data Figure 4g-i.

12. Lipidomic analysis showed associations, but causality is uncertain. Directly manipulating lipids shown to be altered could demonstrate their functional roles.

We thank you for this comment and suggestion. Our lipidomics analysis revealed ferroptosis-related lipid changes, which served as a complementary analysis to support the association of ferroptosis with COVID-19 pathology. In the meantime, we identified changes in other lipid classes, including fatty acids, surfactant lipids and storage lipids. These data are useful for future studies to investigate lipid metabolism involved in inflammation and viral-host interaction. The function and mechanism of specific lipids is not something we were able to manipulate in this current study—we hope you will agree that is outside the scope of what can reasonably be done here.

13. Effects of lipid changes on lung surfactant and pulmonary mechanics were not examined. Biophysical studies on surfactant function would be informative.

We thank you for this comment and suggestion. Our study primarily focuses on ferroptosis mechanism and its association with COVID-19 lung pathology. The function and mechanism of surfactant lipids is useful for future studies of lung surfactant function and pulmonary mechanics, but is not the focus of this current study.

Suggestions for improvement:

14. Increase sample size with age, sex, and comorbidity-matched controls.

Thank you for the suggestion. We have expanded the sample size of both control cases (from 9 to 34) and COVID-19 cases (from 21 to 34), as discussed in previous comment #1. Both sexes were equally involved and control patients were age-matched with COVID-19 patients obtained from the same tissue bank. We chose patient samples based on the pathology of the lung during both gross and microscopic examinations. Control patients used in this study have other medical conditions, but the selected lung regions are unaffected. Comorbidities cannot be matched between control and COVID-19 patients because autopsy samples from healthy subjects were not available. Patient information can be found in Supplementary Data 1.

15. Examine other relevant tissues like the heart, kidney etc.

As discussed in comment #2, we only examined patient lung pathology because the lung is the most affected organ post initial SARS-CoV-2 infection. As suggested, we now examined changes in other organs including heart and kidney from our hamster study post SARS-CoV-2 infection. We did not observe any acute or chronic pathologies in other organs, including the heart (cardiomyocytes, interstitium, and vasculature) and the kidney (glomeruli, tubules, interstitium, and vasculature). These data can be found in Supplementary Figure 2.

16. Conduct additional experiments to clarify the relationship between ferroptosis and lung damage.

We thank you for this comment. As discussed in comment #3, we investigated whether ferroptosis contribute to lung damage by inhibiting ferroptosis using ferroptosis inhibitor, ferrostatin-1 analog Th-2-31 and liproxstatin-1 in hamster model. We found that repeated dosing of Th-2-31 and liproxstatin-1 for 7 and 6 days, respectively, ameliorated lung injury compared to vehicle-treated animals. Ferroptosis marker, TfR1 showed reduction in Th-2-31-treated group compared to vehicle-treated group. This data suggests ferroptosis as a driving force of lung pathology. These data can be found in the result section “Ferroptosis correlates with lung disease severity in a COVID-19 Syrian hamster model” and in Figure 4f, Extended Data Figure 4g-i.

17. Test multiple clinically relevant SARS-CoV-2 strains in vivo.

As discussed in comment #4, we used the wild type strain in the hamster study because it's the same strain found in post-mortem COVID-19 patient lungs used in this study. Therefore, the hamster model recapitulated the human lung disease and was used to study the disease mechanism.

18. Validate lipidomic findings using additional techniques like mass spectrometry imaging.

We thank you for this suggestion. Lipidomics (LC-Ms/MS) provides more accurate ion annotation than mass spectrometry imaging for revealing changes in large number of lipid species, because in imaging mode, we only detect the parent ion, and perhaps use ion mobility. More critically, the lung tissue sections were embedded in OCT, which is not compatible with mass spectrometry imaging, unfortunately. But this is an interesting suggestion for future studies using non-OCT tissue.

19. Assess ferroptosis markers and lipid changes at single cell level.

We thank you for this comment and suggestion. As discussed in comment #9, we re-analyzed published single-cell RNA sequencing dataset (GEO, GSE171524) of COVID-19 lung autopsies obtained from the same cohort of patients used in our study. We find significant changes in the iron regulatory pathway, among which FTL, ferritin heavy chain 1 (FTH1), and TfR1 are the most elevated genes in COVID-19 lung. We further evaluated that iron dysregulation is associated with ferroptosis in the lung. These data can be found in the result section “Dysregulation of iron homeostasis contributes to ferroptosis” and in Figure 2c, Extended Data Figure 2c.

20. Evaluate other cell death pathways besides apoptosis, necroptosis and ferroptosis.

Thank you for this suggestion. As discussed in comment #10, we examined pyroptosis, besides apoptosis, necroptosis, and ferroptosis. We did not observe a significant increase of the pyroptosis marker, cleaved gasdermin D, in COVID-19 lungs. These data can be found in the result section “Ferroptosis is elevated in post-mortem COVID-19 lungs” and Figure 1e.

21. Determine if observed lipid changes impact lung surfactant function.

Thanks for this comment—as discussed in comment #12, our study primarily focuses on ferroptosis mechanism and its association with COVID-19 lung pathology. The function and mechanism of surfactant lipids is useful for future study of lung surfactant function and pulmonary mechanics, but is out of the scope of this current study.

22. Optimize single-cell assays to measure ferroptosis response heterogeneity across lung cell populations.

As discussed in comment #9 and #19, due to the challenges involved in viral inactivation and sample preparation as well as the availability of published single-cell RNA sequencing datasets, we analyzed published dataset [(GEO, GSE171524) of COVID-19 lung autopsies obtained from the same cohort of patients used in our study], particularly focusing on the ferroptosis pathway genes across lung cell populations. We found upregulation of TfR1, FTL, and FTH1 in COVID-19 samples, particularly overexpressed in alveolar macrophages, fibroblasts, and epithelial cells. Additionally, we performed immunofluorescence imaging of ferritin light chain and macrophage markers on COVID-19 autopsy sections and found some FTL signals outside of macrophages. These data suggest that ferroptosis response is heterogenous among different alveolar cell populations. This will facilitate future study on designing targeted therapy to prevent ferroptosis in specific cell types.

23. Assess the biophysical properties of surfactant with lipid composition changes observed in COVID-19.

We thank you for this comment. As discussed in comment #13, our study primarily focuses on ferroptosis mechanism and its association with COVID-19 lung pathology. Assessing the biophysical properties of surfactant is useful for future study of lipid changes on lung function and pulmonary mechanics, but is out of the scope of this current study.

24. Overall, the study provides evidence that ferroptosis contributes to lung injury in COVID-19, but further mechanistic work and validation in additional models would strengthen the conclusions. The suggestions could help elucidate ferroptosis and lipid changes in more detail and assess the therapeutic potential of inhibiting this pathway. If beyond the scope of the manuscript, these limitations should be highlighted and discussed. Finally, this reviewer personally misses some insights and revision of the available literature regarding SARS-CoV-2, WGCNA, bronchial epithelium cells with explainable artificial intelligence and transcriptional profiling can guide novel treatment. The introduction and discussion sections would benefit from this standpoint.

Thanks for this suggestion. As suggested, we have discussed the use of bioinformatic analysis and transcriptomic profiling of COVID-19 lung in facilitating the understanding of pathological processes. The following paragraph can be found in the discussion section:

“Multi-omics profiling and advanced bioinformatic analysis can facilitate the understanding of the molecular basis of pulmonary diseases. Single-nucleus transcriptomic profiling of COVID-19 lungs revealed important pathological hallmarks associated with SARS-CoV-2 infection including hyperinflammation, macrophage infiltration, alveolar epithelial cell exhaustion, parenchymal lung senescence, and fibrosis. A Gene Ontology (GO) and Kyoto Encyclopedia of Genes and Genomes (KEGG) pathway enrichment analysis of published RNA-sequencing dataset revealed ferroptosis-related differentially expressed genes and important biological processes including the oxidative stress response, MAPK, and PI3K-AKT signaling pathway associated with COVID-19. A weighted gene co-expression network analysis (WGCNA) of RNA-sequencing dataset revealed four key genes, PROK2, IL6, TNF, SLC7A11, closely related to COVID-19 ALI. Our analysis of RNA-sequencing dataset (GEO, GSE171524) is consistent with these previous reports, emphasizing the ferroptosis pathway, particularly the iron regulatory pathway associated with severe COVID-19 lungs. Ferroptosis response is heterogeneous among different alveolar cell populations. This will facilitate future studies on designing targeted therapy to prevent ferroptosis in specific cell types.”

Indeed, The mechanistic studies on ferroptosis regulators and lipid enzymes could reveal how SARS-CoV-2 disrupts these pathways to cause lung epithelial cell death. This could uncover new genes and proteins involved in COVID-19 pathogenesis. The animal model experiments testing ferroptosis inhibitors at different time points could help determine if these drugs have potential as COVID-19 treatments and when they would be most effective during illness. Analyzing ferroptosis markers longitudinally in patient samples may identify biomarkers to predict who will develop more severe lung disease. Examining the correlation between specific lipid changes, hub genes/proteins levels, and disease severity could highlight lipid species and genes contributing to pathogenesis. The drug-target network analysis could be expanded by including ferroptosis regulators and lipid enzymes as potential targets. Drugs that modulate these could be candidates. The single-cell assays could characterize if there is heterogeneity in the ferroptosis response across lung cell subpopulations, which may influence susceptibility. Integrating the ferroptosis mechanisms with genomic, transcriptomic and lipidomic analyses could provide a more comprehensive picture of pathogenic pathways in COVID-19. The suggestions could yield new genes of interest, biomarkers, and targeted treatment strategies to improve patient outcomes.

Reviewer #2 (Remarks to the Author):

Qiu et al describe a ferroptosis signature in acute lung injury of COVID19 patients (n=11) as well as Syrian hamster COVID19 model.

Overall, experiments seem to be well performed using different experimental approaches.

Major concerns by this reviewer:

- One can question novelty considering previous reports describing ferroptosis in COVID-19 patients (Refs 18-23).

We thank you for mentioning this. Previous studies unveiled the elevation of serum ferritin and lipid peroxidation in patients but did not evaluate the correlation of ferroptosis with the development of lung pathology. The novelty of this study includes two major aspects: Firstly, we investigated the ferroptosis pathway, particularly the iron metabolism pathway in 3 different systems: patient samples, human lung cells, and the hamster disease model. We found upregulation of iron regulatory proteins including the transferrin receptor 1 (TfR1) and ferritin light chain (FTL) in patient lung samples. Besides, we analyzed a published single-cell RNA sequencing dataset [(GEO, GSE171524) of COVID-19 lung samples obtained from the same cohort of patients used in our study] and found upregulation of TfR1, FTL, and ferritin heavy chain 1 (FTH1) in COVID-19 lung samples, particularly overexpressed in alveolar macrophages, fibroblasts, and other epithelial cells. In addition, we generated an iron-overload model in human primary lung epithelial cells and lung cancer cell line, Calu-1, using ferric ammonium citrate (FAC). FAC treatment induced lipid peroxidation accumulation in primary lung epithelial cells. FAC also induced ferroptotic cell death in ferroptosis-sensitive Calu-1 cells and synergized with ferroptosis inducers RSL3 and IKE in ferroptosis-resistant primary lung epithelial cells. Moreover, we generated a COVID-19 lung disease animal model using Syrian hamsters. We found significantly increased TfR1, FTL, and lipid peroxidation product 4-HNE in hamster lungs at 7 days post SARS-CoV-2 infection. We found that ferroptosis inhibitors reduced lung injury in the hamster model. Altogether, these data confirm that iron overload is associated with ferroptosis in the lung during SARS-CoV-2 infection. These data can be found in the result section “Dysregulation of iron homeostasis contributes to ferroptosis” and Figure 2, Extended Data Figure 2. Secondly, we report a comprehensive analysis of lipid profiles in the lungs of severe COVID-19 cases. This dataset not only supports a ferroptosis lipid signature but also provides insights into the changes of fatty acid, phospholipid metabolism, and surfactant lipid changes that may reveal new targets for preserving lung function.

- A causative role for ferroptosis cannot be claimed with current data set. For this at least, a therapeutic setting using ferroptosis inhibitor (Lip1, UAMC-3203...) should be used in hamster model showing protection against injury.

We thank you for this comment. As suggested, we investigated the efficacy of ferroptosis inhibitor, liproxstatin-1 and ferrostatin-1 analog Th-2-31, in the hamster model. We found that repeated dosing of Th-2-31 and liproxstatin-1 for 7 and 6 days, respectively, alleviated lung injury compared to vehicle-treated animals. Ferroptosis marker, TfR1 showed a reduction in the

inhibitor-treated group compared to the vehicle-treated group. These data suggest that ferroptosis contribute to lung pathology. These data can be found in the result section “Ferroptosis correlates with lung disease severity in a COVID-19 Syrian hamster model” and in Figure 4f, Extended Data Figure 4g-i.

- COVID19 patients are known to be very heterogenous in relation to pathology, comorbidities, medication/treatments... Therefore, a correlation to patient’s clinical data and plasma injury markers is crucial.

We thank you for this comment. We acquired COVID-19 samples from the same cohort of patients that were infected with SARS-CoV-2 (diagnosed in the same timeframe, cause of death by respiratory failure, infected with same strain, stayed in the same hospital) to minimize patient heterogeneity.

- The observed changes in lipid composition fits with what we would expect after ferroptosis cell death wave. However, other biochemical/inflammatory processes can’t be excluded. Therefore, adding an oxidative lipidomics analysis would strengthen the conclusion of the paper.

We thank the reviewer for the perspective on the lipidomics data. We agree that looking at oxidized fatty acyl side chains is valuable to define specific oxidation products that accumulate during ferroptosis or other oxidative stress conditions. However, directly detecting oxidized fatty acyl tails is challenging to perform technically, as the oxidized products are only present in a small percent of the parent lipid species (and can be heterogeneous). Since various oxidation products are typically formed, each is relatively low in abundance and hard to detect. Moreover, defining the identity of specific oxidized lipids is challenging, as it is difficult to identify where the oxidation occurs. We have attempted to do oxidative lipidomics but found it was not reproducible in our hands for the above reasons. We know there are a few labs that can reliably detect oxidized lipids, but they have developed deep expertise and may have specialized instrumentation not available to us. Therefore, while we agree this would be an excellent area of future collaboration, we respectfully propose that this analysis would be beyond the scope of what can be done for this study and is not critical for our conclusions. Although the performance of oxidative lipidomics is not feasible, we examined lipid peroxidation end products, including MDA and 4-HNE using specific antibodies to confirm oxidation of lipids. We also evaluated other cell death markers including for necroptosis and pyroptosis to rule out other inflammatory cell death processes and did not find significant increases in these markers.

Reviewer #3 (Remarks to the Author):

This is an important manuscript that lays the foundation for ferroptosis as a primary driver of cell death in the lung tissue. This is a significant finding that provides a novel target to treat severe ARDS induced by COVID, which is still an unmet clinical need. They convincingly demonstrated ferroptosis in human lung tissue with COVID infections. They also demonstrated that markers of necroptosis and apoptosis were not elevated in the COVID lungs samples. And this was followed by LC-MS lipidomic analysis which showed a depletion of PUFA containing phospholipids and accumulation of lyso-phospholipids which is a signature of ferroptosis driven lipid peroxidation. They confirmed the findings from the human tissues in hamsters infected with

SARS-CoV2. Overall, they provided a convincing case for the discovery of ferroptosis as a major driver of cell death in the lungs of SARS-CoV2 infected cases. All controls and methods were appropriate. I have a few comments and questions below.

The authors did not discuss the role ferroptosis regulators GPX4 or FSP1. It does not appear that GPX4 is expressed in lung tissue can authors comment on what is controlling ferroptosis in lung tissue? or have any evidence for GPX4 or FSP in lung.

We thank you for this comment. As suggested, we investigated the expression of GPX4 and FSP1 by reanalysis of a published single-cell RNA sequencing dataset [(GEO, GSE171524) of COVID-19 lung autopsies obtained from the same cohort of patients used in our study]. We found the expression of GPX4 and FSP1 in alveolar macrophage, fibroblasts, and other epithelial cells. In addition, we detected the expression of GPX4 and FSP1 in patient lung tissues using immunofluorescence and found the expression of both genes in the lung. These data can be found in Extended Data Figure 2c, d.

Method question: The lipid extraction describes extraction from 10 μm sections was it a single section or multiple combined or could they provide an estimate of tissue amount?

We used 20 sections each at 10 μm thickness combined for lipid extraction. We have added this information in the method section.

To inactivate the virus, they incubated the tissue in methanol/acetonitrile mixture which can extract lipids, was this inactivation extract discarded or combined with the Folch extract?

We discarded this inactivation extract. We have added this information in the method section.

Reviewers' Comments:

Reviewer #1:

Remarks to the Author:

The authors have clarified several of the questions I raised in my previous review. Most of the major problems have been addressed by this revision.

Reviewer #2:

None

Reviewer #3:

Remarks to the Author:

Overall, the manuscript is much improved. They have adequately addressed my concerns.

REVIEWERS' COMMENTS

Reviewer #1 (Remarks to the Author):

The authors have clarified several of the questions I raised in my previous review. Most of the major problems have been addressed by this revision.

We appreciate your positive comment and feedback.

Reviewer #3 (Remarks to the Author):

Overall, the manuscript is much improved. They have adequately addressed my concerns.

We appreciate your positive comment and feedback.